# Metabolic and proteomic signatures of type 2 diabetes subtypes in an Arab population

Shaza B. Zaghlool [1], Anna Halama [1], Nisha Stephan[1], Valborg Gudmundsdottir[2,3], Vilmundur Gudnason [2,3], Lori L. Jennings [4], Manonanthini Thangam [5], Emma Ahlqvist [5], Rayaz A. Malik [6], Omar M. E. Albagha [7,8], Abdul Badi Abou-Samra [9] & Karsten Suhre [1] ✉

Type 2 diabetes (T2D) has a heterogeneous etiology influencing its progression, treatment, and complications. A data driven cluster analysis in European individuals with T2D previously identified four subtypes: severe insulin deficient (SIDD), severe insulin resistant (SIRD), mild obesity-related (MOD), and mild age-related (MARD) diabetes. Here, the clustering approach was applied to individuals with T2D from the Qatar Biobank and validated in an independent set. Cluster-specific signatures of circulating metabolites and proteins were established, revealing subtype-specific molecular mechanisms, including activation of the complement system with features of autoimmune diabetes and reduced 1,5-anhydroglucitol in SIDD, impaired insulin signaling in SIRD, and elevated leptin and fatty acid binding protein levels in MOD. The MARD cluster was the healthiest with metabolomic and proteomic profiles most similar to the controls. We have translated the T2D subtypes to an Arab population and identified distinct molecular signatures to further our understanding of the etiology of these subtypes.

Type 2 diabetes (T2D) is a complex metabolic disorder defined by dysregulated glucose homeostasis, driven by imbalanced energy intake and expenditure, dysfunction of insulin signaling, and chronic inflammation[1–3]. Multiple therapies are now available to improve glycemic control[4] and provide additional benefits in relation to complications[5–7]. Indeed, individualized therapies targeting the underlying pathophysiology and complications should be a major goal in the treatment of patients with T2D[1].

Diabetes is presently classified into type 1 and type 2 diabetes. Ahlqvist et al.[8] used data on age at diagnosis, BMI, HbA1c, homeostasis model assessment (HOMA) estimates of beta-cell function (HOMA2-B), insulin resistance (HOMA2-IR), and glutamic acid decarboxylase antibodies (GADA) to stratify subjects into four clusters representing T2D subtypes (SIDD, SIRD, MOD, and MARD) and one cluster with severe autoimmune diabetes (SAID), which represents type 1 diabetes. We follow the same stratification for those with type 2 diabetes, however, for those with type 1 diabetes we use C-peptide levels rather than GADA[8]. The four T2D clusters were named in reference to their characterizing phenotypic signatures as Severe Insulin Deficient Diabetes (SIDD), Severe Insulin Resistant Diabetes (SIRD), Mild Obesity-related Diabetes (MOD), and Mild Age-related Diabetes (MARD). Since its publication in 2018, the paper has been cited over 1300 times and discussed in multiple reviews[1,9–13]. The clusters have been replicated in British[14], German[15,16], Mexican American and Chinese[17–19], Japanese[20], Asian Indian[21], Mexican[22], and Icelandic[23] cohorts, suggesting generalizability to other ethnicities. In the original analysis, predisposition to retinopathy and nephropathy were identified in different clusters, and more recently a

[1]Department of Physiology and Biophysics, Weill Cornell Medicine-Qatar, Doha, Qatar. [2]Faculty of Medicine, University of Iceland, Reykjavik, Iceland. [3]Icelandic Heart Association, Kopavogur, Iceland. [4]Novartis Institutes for Biomedical Research, Cambridge, MA, USA. [5]Department of Clinical Sciences Malmö, Lund University, Lund, Sweden. [6]Weill Cornell Medicine-Qatar, Doha, Qatar. [7]College of Health and Life Sciences, Hamad Bin Khalifa University, Education City, Doha, Qatar. [8]Centre for Genomic and Experimental Medicine, Institute of Genetics and Cancer, University of Edinburgh, Edinburgh, UK. [9]Qatar Metabolic Institute, Hamad Medical Corporation, Doha, Qatar. ✉e-mail: kas2049@qatar-med.cornell.edu

German cohort cluster analysis has revealed predisposition to non-alcoholic fatty liver disease and diabetic neuropathy[15]. A study clustering genetic risk loci for T2D-associated traits observed some overlap with the clusters of Ahlqvist et al.[24]. More recently, evidence for distinct genetic backgrounds of the subtypes has been found[25]. Schüssler-Fiorenza Rose et al.[26] used multi-omics measurements in a longitudinal study to develop prediction models for insulin resistance and the German Diabetes Study (GDS) showed differences in protein biomarkers of inflammation between subgroups[16]. These studies suggest that deep molecular phenotyping may provide key insights into the underlying pathophysiology of glucose dysregulation and the development and progression of comorbidities in patients with T2D.

Whilst a mechanistic link has been suggested between increased fat storage and compromised glucose homeostasis[27], body mass index (BMI) alone does not explain the difference between normal and dysfunctional glucose metabolism. Specific metabolic and proteomic processes may help to characterize the broader spectrum of physiological perturbations associated with impaired glucose metabolism to enable subtype-specific individualization of therapies. We hypothesized that there are distinctive alterations in metabolic and proteomic components of signaling pathways underlying the different T2D clusters.

We have analyzed data from the Qatar Biobank (QBB) population and translated the Ahlqvist et al. clustering approach to an Arab population. Further, applying broad non-targeted metabolomics and affinity proteomics profiling we have identified cluster-specific physiological and biochemical processes in relation to their predominant treatment regimens. The complete study design is presented in Fig. 1.

## Results

### The T2D subtype clustering scheme defined for Caucasians can be translated to an Arab population

Following the approach of Ahlqvist et al., we used k-means clustering of age at diagnosis, BMI, HbA$_{1c}$, HOMA2-B, and HOMA2-IR and identified four clusters with clinical properties similar to those in the ANDIS study (Fig. 2A). The SIDD cluster was characterized by young age at onset, low BMI, low insulin secretion (HOMA2-B) and poor glycemic control (high HbA$_{1c}$); the SIRD cluster had the highest level of insulin resistance (HOMA2-IR) and high BMI; the MOD cluster had a high BMI with low insulin resistance; and the MARD cluster, like the MOD clusters, had low insulin resistance, but a much lower age of onset of T2D. The relative cluster sizes in QBB were comparable to those found in the ANDIS study, except for SIRD, which made up only 4% of the T2D cases in QBB compared to 15% in ANDIS.

We performed several sensitivity tests on the way the clusters were derived. First, we replicated the clustering in an independent testing set within QBB. We obtained very similar results compared to using cluster coordinates from the training set (Fig. 2B) and found that cluster assignments were identical for 98% of the study participants. We then repeated the clustering allowing for a varying number of clusters (Supplementary Fig. 1A). Consistent with the observations of Ahlqvist et al., four clusters were identified in QBB. Allowing for a fifth cluster led to a split of the MARD cluster into one cluster with a lower and one with a higher age of T2D onset (Supplementary Fig. 1A). Repeating the cluster analysis separately for females and males showed that most individuals (93%) were assigned to the same cluster as in the initial analysis (Supplementary Fig. 1B). Although there was a slight imbalance between males and females in the clusters (51 vs. 42 in SIDD, 11 vs. 6 in SIRD, 23 vs.

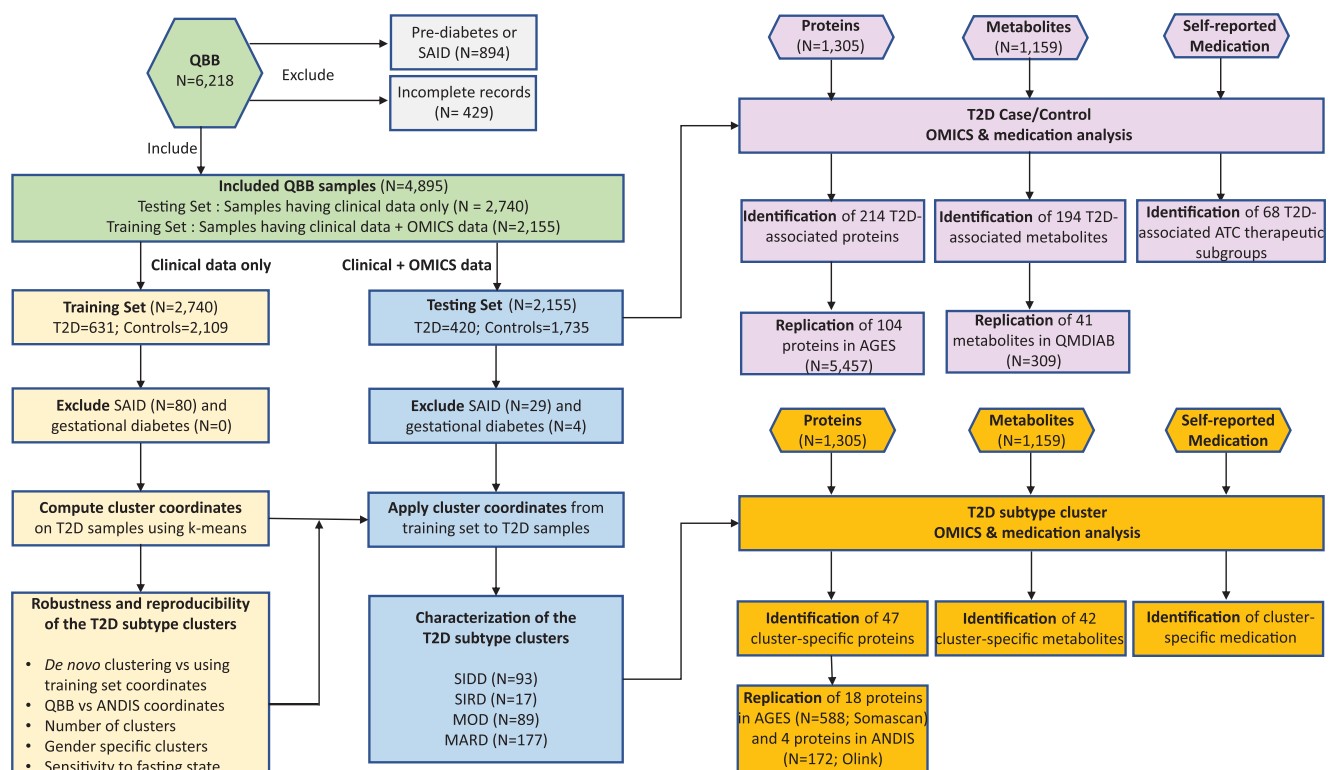

**Fig. 1 | Study design.** The study includes two main parts. The first part consists of the identification of T2D subtypes in an Arab population by applying a clustering scheme from the ANDIS study, comparing the Arab clusters to the European clusters, and testing for various confounding factors. The second part includes the omics (proteins and metabolites) and medication analysis in both the case/control setting and the T2D subtype-specific setting. T2D type 2 diabetes, ANDIS All New Diabetics in Scania, SAID severe autoimmune diabetes, SIDD severe insulin-deficient diabetes, SIRD severe insulin-resistant diabetes, MOD mild obesity-related diabetes, MARD mild age-related diabetes.

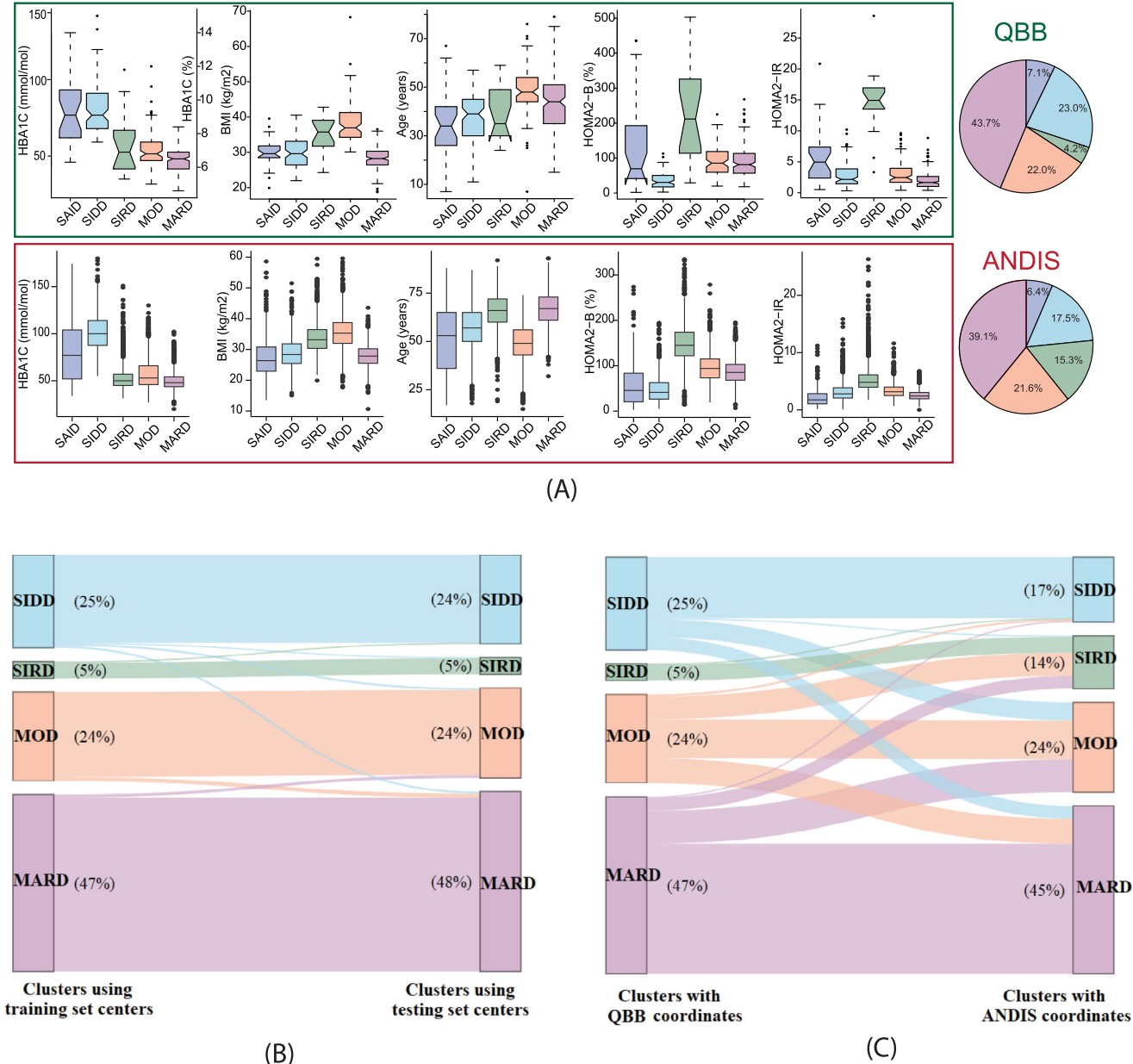

**Fig. 2 | Cluster characteristics and cluster distribution in QBB.** K-means clusters were derived using the QBB training set and classification was applied to the testing set using the training set cluster coordinates. **A** Distributions of HbA$_{1c}$, BMI, age, HOMA2-B, and HOMA2-IR are shown for each cluster in QBB ($N$ = 420 individuals) and ANDIS ($N$ = 8980 individuals). HbA$_{1c}$, BMI, HOMA2-B, and HOMA2-IR all followed the same trend in QBB and ANDIS, but individuals in the MOD cluster were younger than the other clusters in QBB. Data in boxplots are presented as follows: lower and upper whiskers represent the minima and maxima respectively, box centers represent the median values, bounds of boxes represent the first and third quartiles, notches represent the 95% confidence interval of the median, and circles represent outliers. **B** The testing set clusters were similar to the training set clusters, regardless of whether they were assigned based on the training set coordinate centers or derived de novo for the testing set using K-means clustering. Minor

changes in the cluster assignments (2%) were observed when clustering the data using the training set coordinates versus the testing set coordinates. **C** The ANDIS coordinates were used instead of the QBB coordinates to classify QBB patients. Gender-specific type 2 diabetes cluster centers (SIDD, SIRD, MOD, and MARD) from Ahlqvist et al.[8] were obtained. After computing the Euclidean distance between the four clusters and each individual in QBB, each individual was assigned to the cluster with the shortest distance. When comparing the cluster assignment that was based on the QBB coordinates vs. ANDIS coordinates, a 35% change in the cluster assignments was observed. QBB: Qatar Biobank, ANDIS: All New Diabetics in Scania, SAID: severe autoimmune diabetes, SIDD: Severe Insulin Deficient Diabetes, SIRD: Severe Insulin Resistant Diabetes, MOD: Mild Obesity-related Diabetes, MARD: Mild Age-related Diabetes.

66 in MOD, and 102 vs. 75 in MARD), clustering the males and females separately (Supplementary Fig. 2) resulted in little difference in the eventual cluster membership. The 26 (7%) individuals on the borderline of the different clusters included 11 males and 15 females. Due to such small numbers, it is difficult to judge whether the individuals on the borderline of the different cluster groups were more likely to be one sex.

HOMA2-IR and HOMA2-B estimates are based on plasma glucose and C-peptide (Cpep) levels and are sensitive to the fasting state. Other studies have reported on clustering using non-fasting values by using HDL-cholesterol and C-peptide, which is a proxy for insulin resistance[28]. As QBB participants were not all in a fully fasting state (77% of the individuals had fasted for over two hours at the time of enrollment and 50.7% had fasted for over 8 h), we tested the

sensitivity of the cluster assignments to self-reported time-since-last-meal. Using linear regression, we estimated fasting HOMA2-IR and HOMA2-B and used the corrected values for clustering. 98% of the cluster assignments remained unchanged, indicating that the clustering is robust to fasting state (Supplementary Fig. 1C).

Finally, we tested whether ANDIS-derived cluster centers could be used directly to classify QBB participants into T2D subtype clusters. We observed consistent cluster assignments for 65% of the individuals when using the ANDIS-derived cluster centers instead of the QBB-derived cluster centers for classification (Fig. 2C). In addition, we noted a gender bias in the misclassification with 23% of misclassified males compared to 46% of females. A comparison of the cohort-specific cluster centers (standardized values centered to mean = 0 and SD = 1) is presented in Supplementary Data 3. Apart, from the age variable, gender-specific variables for the different clusters were directionally consistent between ANDIS and QBB (Supplementary Fig. 3). Although the trends of four out of five cluster variables were consistent across the T2D subtypes in both cohorts, the limited overall agreement between T2D subtype classifications obtained using ANDIS versus QBB coordinates suggests that population specific coordinates should be used. We therefore used sex-independent coordinates derived from the QBB training set to classify the QBB training set in the following metabolomics and proteomics analyses.

## Metabolomics and proteomics associations with T2D replicate in other populations

Diabetes-specific alterations of protein and metabolite levels have been previously described in different populations[29–32]. To validate the metabolomic and proteomic data in QBB and its ability to characterize T2D participants, we investigated omics associations with T2D. Deep molecular phenotyping data was available for 420 QBB participants with T2D and 1735 controls and covered semi-quantitative measures of 1159 blood circulating metabolites and 1305 plasma proteins. We compared the protein levels of the T2D cases ($N = 420$) to those of the controls ($N = 1735$) in a linear model with covariates as described in the methods. We identified 214 proteins associated with T2D at a Bonferroni level of significance ($p < 0.05/1305 = 3.83 \times 10^{-5}$) (Fig. 3A and Supplementary Data 4). We checked the associations for replication in the independent European AGES population ($N = 5457$) with SOMAscan protein measurements. Of 214 proteins associated with T2D in QBB, 107 were also reported as associated with T2D in AGES at a study-wide significance threshold ($p < 0.05/4,782 = 1.04 \times 10^{-5}$)[23]. One hundred and four (97.2%) of these proteins replicated at a Bonferroni-corrected significance level ($p < 0.05/107 = 4.67 \times 10^{-4}$). All replicated associations were directionally concordant (Fig. 3B). Variable transformations were chosen to be compatible with the replication cohorts (Box-Cox for AGES, log for QMDiab). We verified that using a log transformation did not substantially change the results.

We further identified 194 metabolites that were associated with T2D ($p < 0.05/1159 = 4.31 \times 10^{-5}$) (Fig. 3C & Supplementary Data 5). We confirmed previous T2D associations with sugars (glucose, mannose, 1,5-anhydroglucitol (1,5-AG), etc.), with branched-chain amino acids (BCAAs) (incl. isoleucine, leucine, valine), and various lipids and markers of kidney function. We attempted replication of these T2D-metabolite associations in the multi-ethnic Qatar Metabolomics Study of Diabetes (QMDiab) cohort (Supplementary Data 6). From the 194 T2D-associated metabolites identified in QBB, 175 were also measured in QMDiab. All associations were directionally concordant. Despite being less powered, 41 (23%) of these associations were statistically significant at a Bonferroni level ($p < 0.05/175 = 2.85 \times 10^{-4}$) in the QMDiab study (Fig. 3D).

## Cluster-specific metabolomics and proteomics associations reveal diabetes subtype-specific processes

We identified all proteins and metabolites that were differentially expressed in one of the four T2D subtype clusters. We required (1) that their means were different from those of all other clusters combined at a Bonferroni significance level and (2) that their means were different from all other clusters in a pair-wise comparison at a nominal level of significance (see methods). Based on this criterion, 47 proteins and 42 metabolites were specific to a given T2D subtype. Figures 4 and 5 represent an overview of the central findings (data in Supplementary Data 7 and Supplementary Data 8, Fig. 6, and Supplementary Data 1 and Supplementary Data 2 have detailed boxplots). In the following, we report highlights of these associations and possible rationalizations for the observed subtype specificities. We start with proteins, followed by metabolites, and address which are—in our view—the most interesting findings, always following the same order, that is, SIDD, SIRD, MOD, and then MARD.

We observed subtype-specific elevated levels of Complement C2 (C2) in the SIDD cluster. Type 1 diabetes has a well-established association with HLA antigens[33] and C2 has been linked to HLA in type 1 diabetes[34]. SIDD is most similar to SAID in terms of being the most severe insulin deficient, but is not auto-antibody positive, which is the primary diagnostic feature in SAID. Complement activation extends beyond microbial defense and can be involved in obesity, insulin resistance, diabetes, and dyslipidemia, indicating an inflammatory component[35–39]. Studies have shown that metabolic inflammatory signaling can affect pathways that lead to insulin resistance[40,41]. Accumulating evidence supports the activation of the complement system with the development of insulin resistance[42]. The SIRD cluster had the highest levels of insulin (INS) and the lowest levels of insulin-like growth factor-binding protein-1 (IGFBP1). Individuals could develop insulin resistance due to low IGFBP1 which directly affects insulin sensitivity through its RGD domain[43]. Proteins (C59 glycoprotein, inhibin beta A chain, osteomodulin, Follistatin-related protein 3, C27 antigen) specifically dysregulated in SIDD were often also dysregulated in SAID, possibly reflecting shared underlying processes. The MOD cluster had the highest leptin (LEP) levels and enzymes involved in lipid metabolism, such as phospholipase A2 (PLA2G2A) and fatty acid-binding protein (FABP3). The MARD cluster had the highest levels of APOM, APOB, UNC5D, NCAM1, Cystatin-M, and the lowest levels of Plexin-B2 (PLXNB2). All protein levels specific to MARD were closer (or comparable) to those of the controls when compared to the other subtypes, suggesting that MARD individuals were the healthiest among the T2D subtypes, and that the proteins associated with MARD were more strongly dysregulated in the other subtypes.

In relation to plasma metabolites, individuals in the SIDD cluster had the lowest 1,5-AG levels of all groups. 1,5-AG is a marker of short-term glycemic control and is implemented as a clinical test in the GlycoMark™ assay[44]. The blood sugars mannose, glucose, fructose, mannonate, and gluconate were considerably higher in the SIDD cluster, indicating a greater level of hyperglycemia. Individuals in the SIDD cluster also exhibited elevated levels of cortisone and cortisol, which are stress markers associated with dysregulated glucose metabolism[45] and a number of chronic complications of T2D[46]. We also observed a decrease in the level of dimethylglycine (DMG) in SIDD, a product of betaine catabolism and low betaine and DMG levels, which has been associated with higher glucose levels and the development of T2D[47]. Furthermore, the SIDD cluster had decreased levels of gamma-glutamyl amino acids (gamma-glutamylphenylalanine and gamma-glutamyltyrosine), indicating perturbed glutathione metabolism. The levels of two sphingomyelin species (sphingomyelin (d18:2/14:0, d18:1/14:1) and sphingomyelin (d18:2/24:2)) were also lower in the SIDD cluster. Downregulated sphingolipid metabolism can affect insulin sensitivity and lead to β cell dysfunction[48].

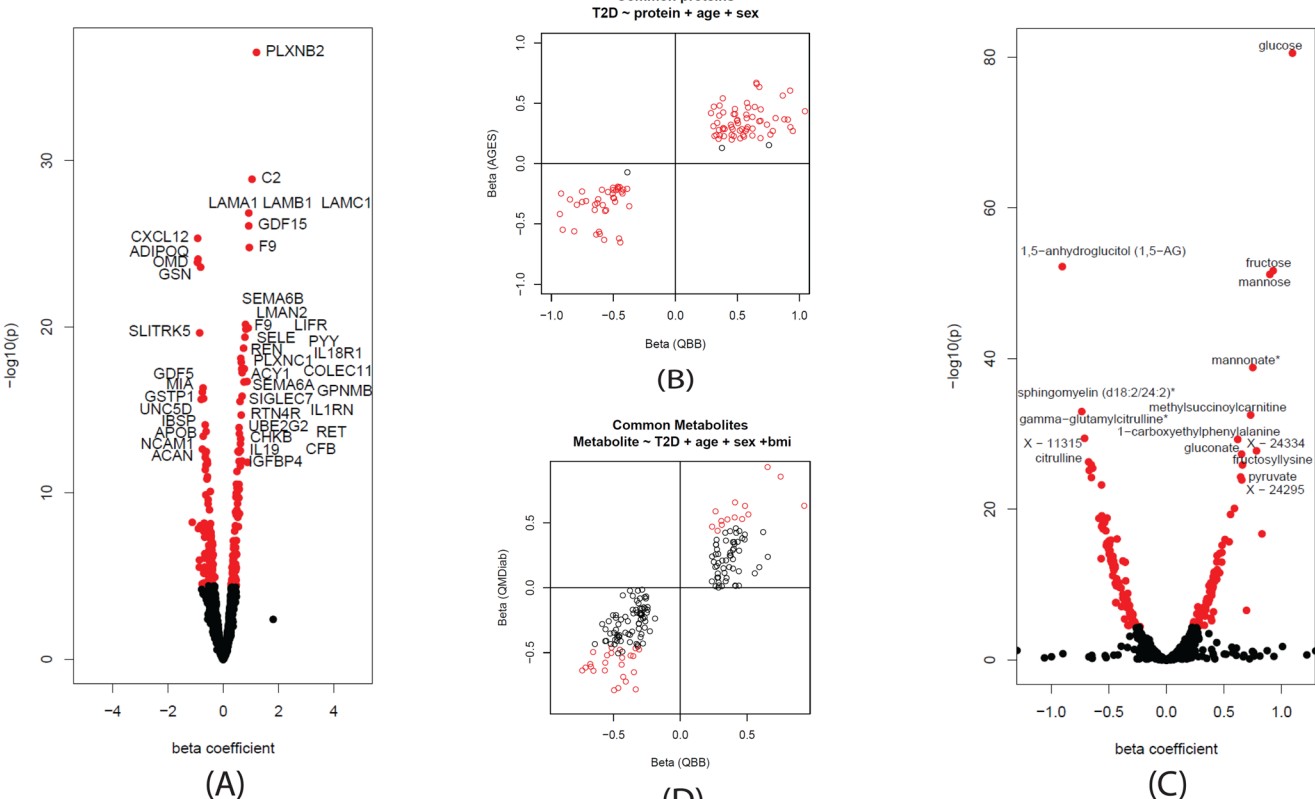

**Fig. 3 | Associations of proteomics and metabolomics levels with T2D. A** Using a linear model, 214 proteins were significantly associated at a Bonferroni level of significance ($p < 0.05/1305 = 3.83 \times 10^{-5}$) with T2D in QBB after adjusting for age and sex. **B** Comparison of effect sizes between QBB and AGES. The replication status of 107 common proteins is shown red: Bonferroni significant ($p < 0.05/107 = 4.67 \times 10^{-4}$) in both studies, black: significant only in QBB. **C** Using a linear model, 194 metabolites were significantly associated at a Bonferroni level of significance ($p < 0.05/1159 = 4.31 \times 10^{-5}$) with T2D after adjusting for age, sex, and BMI. **D** Comparison between QBB and QMDiab. The replication status of the 175 common metabolites is shown in red/black–Bonferroni significant ($p < 0.05/175 = 2.86 \times 10^{-4}$) or significant only in QBB. T2D: Type 2 Diabetes, QBB: Qatar Biobank, QMDiab: Qatar Metabolomics Study of Diabetes.

The SIRD cluster had elevated levels of lipokine-related metabolites, including 12,13-DiHOME, a linoleic acid metabolite, and 2-hydroxyarachidate, an arachidic acid metabolite. 12,13-DiHOME was previously recognized as an important lipid mediator stimulating fatty acid uptake by skeletal muscles[49], which could serve as an alternative energy source for SIRD subjects given their potentially limited access to glucose, due to insulin resistance. The plasma level of phosphate was lower in the SIRD cluster compared to other clusters, and could potentially be linked to hypophosphatemia which is frequently observed in conditions of diabetic ketoacidosis driven by hyperglycemia-induced osmotic diuresis[50]. Although diabetic ketoacidosis is rare in SIRD, it has been previously reported[51].

The MOD cluster had high levels of hydroxyasparagine, 5−(galactosylhydroxy)−L−lysine, and 7-alpha-hydroxy-3-oxo-4-cholestenoate (7−Hoca), which play a role in lipid metabolism and obesity[52]. The elevated levels of 5−(galactosylhydroxy)−L−lysine, an important post-translationally modified amino acid present in collagen-like proteins, such as adiponectin[53], may be the result of adipogenic collagen turnover[54,55]. Metalloproteinase, MT1-MMP, a pericellular collagenase and a member of the matrix metalloproteinase (MMP) gene family, directs interactions that control adipogenesis[54] and is critical to white adipose tissue development by remodeling the 3-D type I collagen scaffolding that dominates primordial white fat deposits. Hence, its absence leads to disruption of transcription factor cascades required for adipocyte maturation and would broadly occur in individuals in the MOD cluster with high BMI. Adiponectin is an important target in obesity treatment, is a key regulator of fatty acid oxidation and lipid synthesis, and is well known to decrease triglyceride concentrations

and increase insulin sensitivity[56]. Oxysterols play a signaling role in lipid and glucose metabolism which may be implicated in obesity through the control of lipogenesis[57,58]. They also play an important role in cholesterol uptake, transport, excretion, and gene regulation[59–61]. The elevated levels of oxysterol, 7-Hoca, may be a result of dysregulated fatty acid metabolism and lipid homeostasis.

As in the case of proteins, the metabolic profiles of individuals in the MARD cluster were closest to the controls. Blood carbohydrates levels (glucose and fructose) were higher than normal but were the lowest among the T2D clusters. The levels of glycine, glutamine, histidine, and gamma-glutamyl amino acids (gamma-glutamylglycine, gamma-glutamylglutamine, and gamma-glutamylthreonine) were the lowest in MARD and comparable to individuals without diabetes. Glycine and glutamine are both implicated in insulin secretion[62,63]. Glycine acts on the pancreas through glycine receptors and as a co-ligand for N-methyl-d-aspartate glutamate receptors to control insulin secretion and glutamine regulates beta-cell gene expression, signaling, and insulin secretion. In addition, histidine and gamma-glutamyl amino acids play a role in anti-inflammatory and antioxidative responses[64,65]. Histidine supplementation has been shown to improve insulin resistance by suppressing pro-inflammatory cytokine expression, possibly through the nuclear factor kappa-B (NF-κB) pathway[64]. Serum gamma-glutamyltransferase is strongly linked to obesity and non-alcoholic fatty liver disease, which may lead to systemic and hepatic insulin resistance, respectively[66]. Overall, the MARD patients had the least metabolic dysregulation among the T2D subtypes.

We computed the explained variance of all proteomic and metabolomic principal components (PCs), by cluster membership, using a

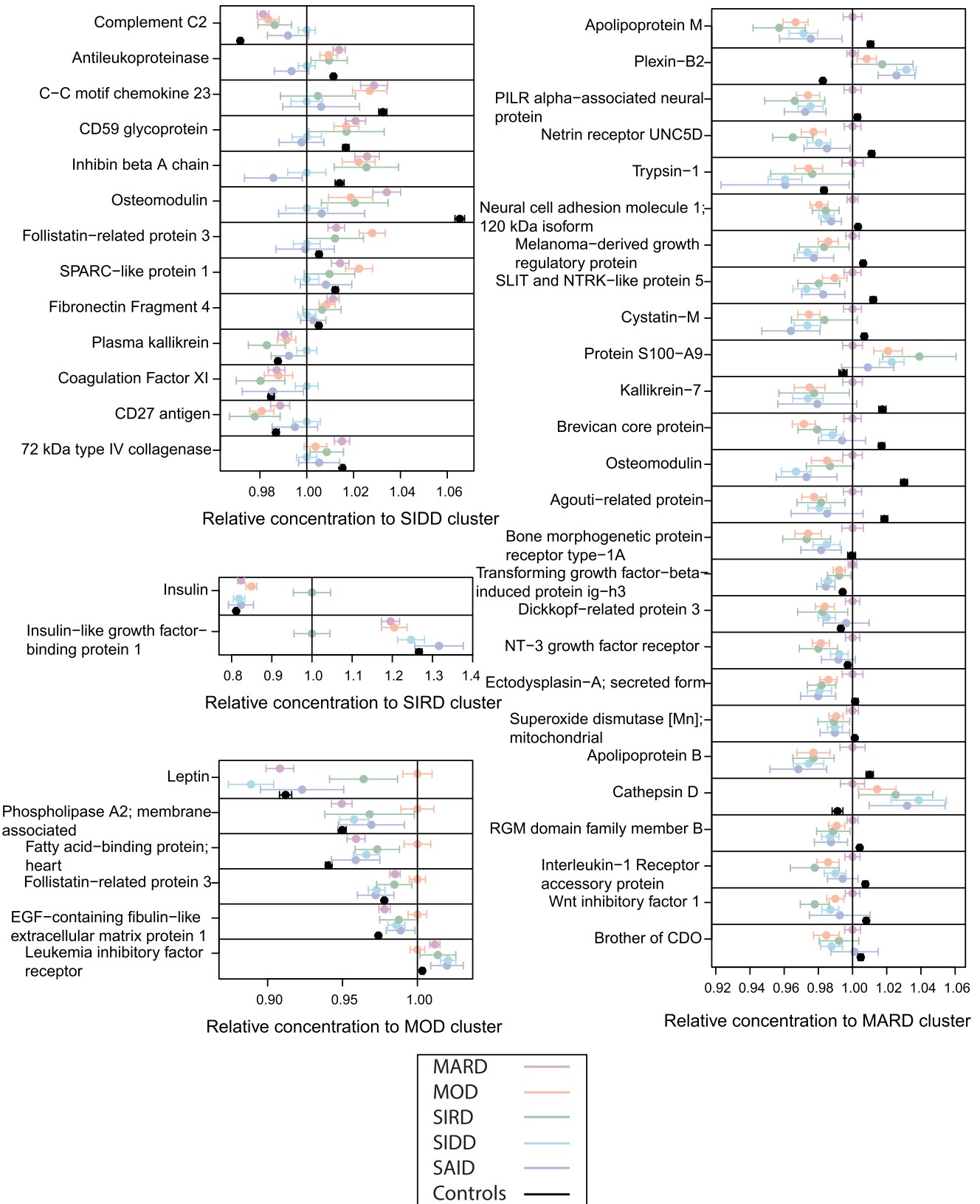

**Fig. 4 | Proteins that distinguish individual diabetes subtypes (*N* = 420 individuals).** The dots and bars represent the mean protein values and the 95% confidence intervals of the means for proteins that are different in one of the four T2D subtypes compared to all others. Values are normalized by the mean of the respective reference subtype. In addition, data for SAID and the control group are shown for reference. T2D type 2 diabetes, SAID severe autoimmune diabetes, SIDD severe insulin-deficient diabetes, SIRD severe insulin-resistant diabetes, MOD mild obesity-related diabetes, MARD mild age-related diabetes.

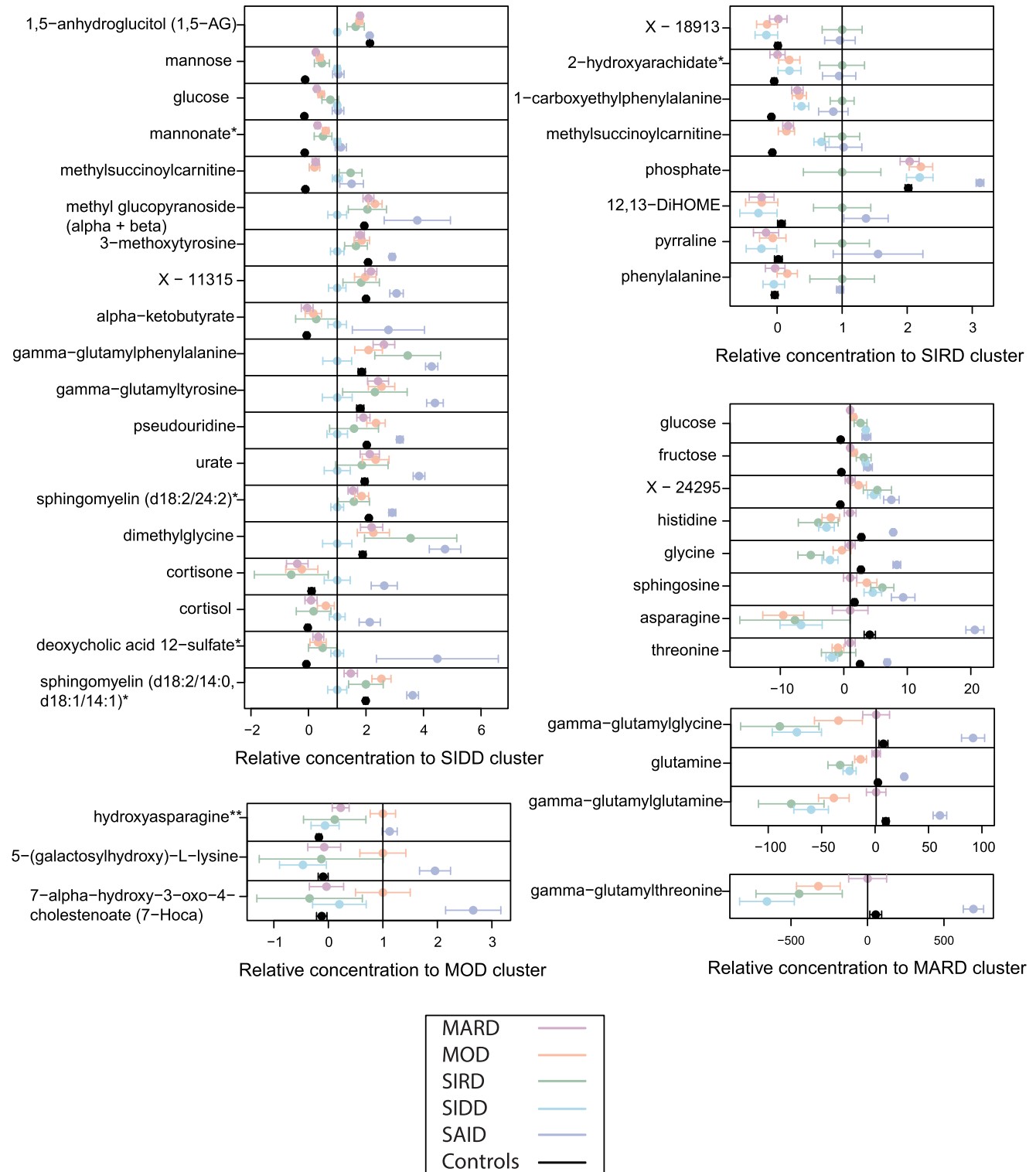

**Fig. 5 | Metabolites that distinguish individual diabetes subtypes (*N* = 420 individuals).** The dots and bars represent the mean metabolite values and the 95% confidence intervals of the means for metabolites that are different in one of the four T2D subtypes compared to all others. Values are normalized by the mean of the respective reference cluster. In addition, SAID and the control group is shown for reference. T2D type 2 diabetes, SAID severe autoimmune diabetes, SIDD severe insulin-deficient diabetes, SIRD severe insulin resistant diabetes, MOD mild obesity-related diabetes, MARD mild age-related diabetes.

linear regression model. Proteomics PC2, PC7, and PC4 yielded the highest R2 of 15.5%, 14.3%, and 10.0%, respectively. Similarly, the cluster membership explained 18.2%, 7.5%, and 7.5% of the variance in metabolomics PC3, PC7, and PC9, respectively. However, as the interpretation of PCA analysis is challenging, we did not seek further explanations in this study.

To combine the results from the metabolomics and proteomics datasets, we conducted a correlation analysis between the cluster-specific proteins and metabolites (Supplementary Figs. 4–6). Some correlation can be observed between the proteins and metabolites, some of which are within the same subtype, particularly in the MOD subtype, while others are between different subtypes. These results are

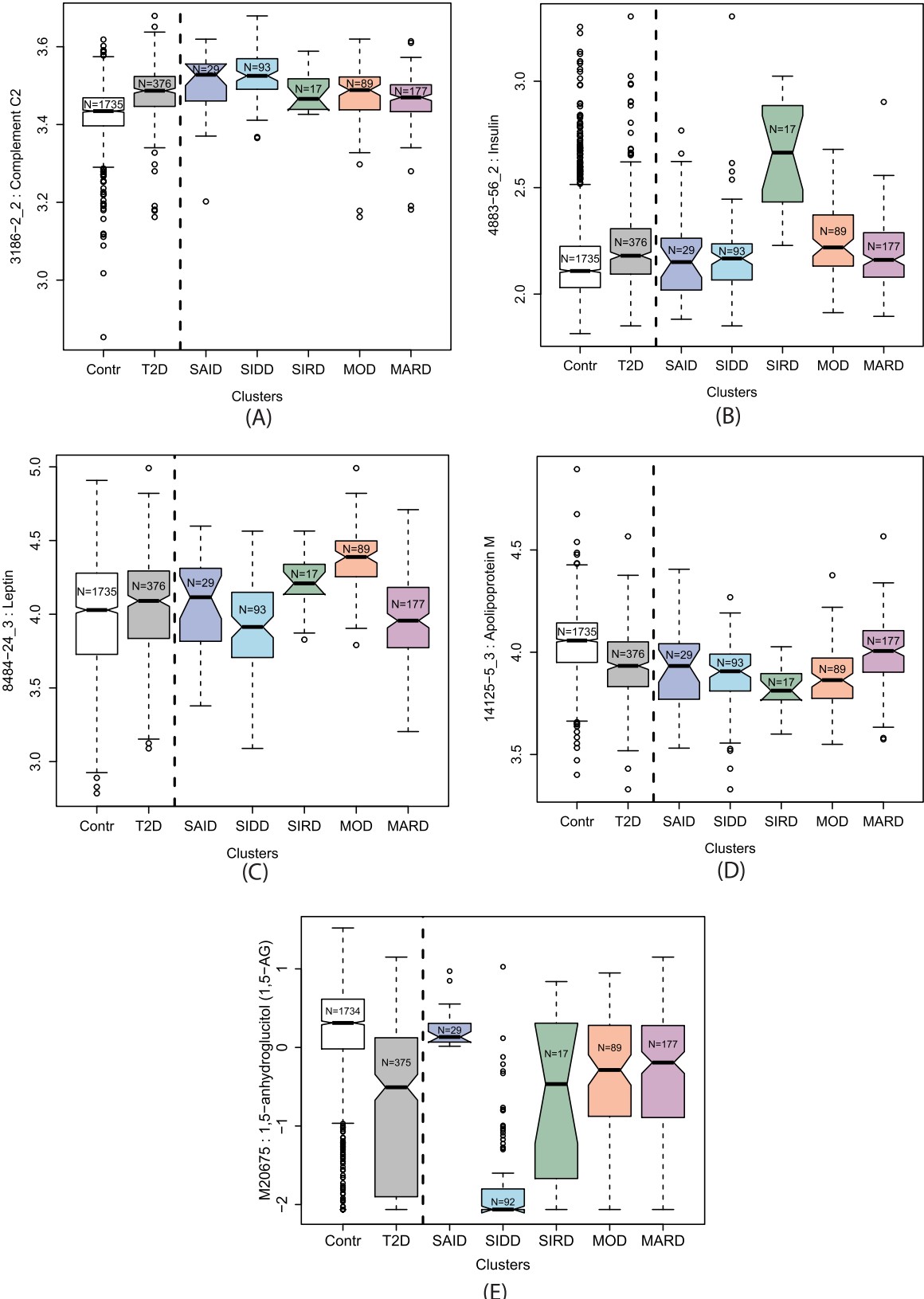

**Fig. 6 | Boxplots of the significantly altered proteins or metabolites in specific diabetes subtypes. A** Complement C2 (C2), **B** insulin (INS), **C** leptin (LEP), and **D** Apolipoprotein M (APOM) are significantly higher in SIDD, SIRD, MOD, and MARD, respectively, while **E** 1,5-AG is significantly lower in SIDD. The complete set of boxplots for all protein and metabolite levels for each cluster are in Supplementary Data 1 and Supplementary Data 2. Contr Controls, T2D type 2 diabetes,

SAID severe autoimmune diabetes, SIDD severe insulin-deficient diabetes, SIRD severe insulin-resistant diabetes, MOD mild obesity-related diabetes, MARD mild age-related diabetes. Data in boxplots are presented as follows: lower and upper whiskers represent the minima and maxima respectively, box centers represent the median values, bounds of boxes represent the first and third quartiles, notches represent the 95% confidence interval of the median, and circles represent outliers.

not easy to interpret due to confounding factors such as sex, age, etc. which may be driving some of these cross-subtype associations. Further analysis is beyond the scope of this study.

### Assessment of the effect of confounding on cluster membership and individual metabolites and proteins

We tested for the association of the following confounding factors: sex, time-since diagnosis, fasting time, medication (lipid lowering, diabetes, and hypertension drugs), and T2D polygenic risk score with the cluster assignment. The score for T2D was computed using variants and weights from a previous study[67] based on summary statistics from a recent GWAS in patients with type 2 diabetes. After correcting the five clustering variables for the significant confounders (sex, time-since diagnosis, diabetes drugs, and hypertension drugs) all associations remained significant ($p < 0.05$). We further independently reported potential confounding of the various factors with the identified cluster-specific proteins and metabolites (Supplementary Data 9 and Supplementary Data 10). It appears that the time-since disease diagnosis is higher in SIDD compared to the other subtypes, suggesting that the more severe subtypes of T2D may possibly be the result of disease progression. However, it is also reasonable that the more severe subtypes have been diagnosed for a longer period.

We further tested for the effects of diet and microbiome on cluster outcomes. Using data from a previous study[68] which identified 335 metabolites that were significantly explained by diet and 182 by the microbiome. We extracted the explained variance of diet and microbiome ($R^2$) by our cluster-specific metabolites (Supplementary Data 11). However, we found no strong correlation between these metabolites and microbiome ($R^2 < 0.15$) nor diet ($R^2 < 0.35$).

### Replication of T2D subtype-specific protein associations in a different population using the same proteomics platform

We attempted replication of T2D subtype-specific protein associations in a different population using the same proteomics platform. This was attempted for all 47 subtype-specific protein associations (Supplementary Data 12; Supplementary Fig. 7) using proteomics measurements from the Somalogic platform in the AGES study[23]. Data was available for 588 individuals (SIDD = 61, SIRD = 84, MOD = 120, MARD = 212). Here, 18 protein associations were replicated after accounting for multiple testing ($p < 0.05/47 = 0.00106$). Additionally, 16 protein associations were nominally significant ($p < 0.05$) and directionally consistent in AGES. Furthermore, 96% of the associations were directionally consistent between the two cohorts (Supplementary Fig. 8).

Slieker et al.[69] also analyzed the molecular signatures of the diabetes subtypes after obtaining metabolomic, lipidomic, and proteomic data from plasma. Molecular measures were measured using ultrahigh-performance liquid chromatography-tandem mass spectrometry (UHLPC-MS/MS) for metabolomics ($N = 15$ metabolites), and the SomaLogic somascan platform for proteomics ($N = 1195$ proteins). We tested in QBB the 906 protein measures reported by Slieker et al. to be specific for the SIDD, SIRD, MOD, and MARD clusters, and found that 38%, 2%, 19%, and 33% of the proteins replicated respectively, while 75%, 50%, 57%, 66% of the proteins had consistent directionality, respectively (Supplementary Data 13).

### Replication of T2D subtype-specific molecular associations in a different population using a different platform

We attempted replication of 30 out of 47 subtype-specific protein associations (Supplementary Data 14) using proteomics measurements from the Olink platform in ANDIS. Data were available for 43 individuals in each of the four T2D subgroups ($N = 172$). Four protein associations were replicated after accounting for multiple testing ($p < 0.05/30 = 0.00167$). These were Follistatin-related protein 3 (FSTL3) with the SIDD cluster, Plexin-B2 (PLXNB2) and Cathepsin D

(CTSD) with the MARD cluster, and leptin (LEP) with the MOD cluster. Five additional proteins showed concordant directionality at a nominal significance level ($p < 0.05$), that is, CD59 glycoprotein (CD59), Complement C2 (C2), and 72 kDa type IV collagenase (MMP-2) with SIDD, Leukemia inhibitory factor receptor (LIF-4) with MOD, and NT-3 growth factor receptor (NTRK-3) with MARD.

We further attempted to replicate 14 of the 15 measured metabolic associations reported in Slieker et al.[69] (measured using UHLPC-MS/MS). Here, 50%, 25%, 0%, and 67% of the metabolites replicated in the SIDD, SIRD, MOD, and MARD clusters respectively, while 100%, 88%, 0%, and 100% of the metabolites had consistent directionality, respectively (Supplementary Data 15).

### Medication patterns are cluster specific

Self-reported drug usage was annotated using unique active molecule identifiers obtained from the Drugbank repository[70] and their corresponding ATC codes. We compared difference in medication usage from all ATC anatomical groups between T2D cases and controls using a fisher test (Supplementary Data 16). The most common drugs used by patients with T2D were in the ATC anatomical main groups A (Alimentary tract and metabolism) and C (Cardiovascular system). In descending order, the most frequently administered drug subgroups by T2D subjects were A10: Drugs used in diabetes ($p = 8.70 \times 10^{-220}$), C10: lipid modifying agents ($p = 1.04 \times 10^{-64}$), B01: anti-thrombotic agents ($p = 2.43 \times 10^{-52}$), C07: beta blocking agents ($p = 1.74 \times 10^{-51}$), A01: stomatological preparations ($p = 7.51 \times 10^{-41}$), N02: analgesics ($p = 1.17 \times 10^{-39}$), M01: anti-inflammatory and anti-rheumatic products ($p = 9.40 \times 10^{-32}$), and C09: agents acting on the renin-angiotensin system ($p = 3.40 \times 10^{-30}$) (Supplementary Data 17).

The medication patterns were also compared across the diabetes subtype clusters in QBB (Supplementary Data 18). Patients in the SIDD cluster were more frequently using insulin ($p$ value = $2.62 \times 10^{-4}$), or Metformin ($p = 1.12 \times 10^{-3}$), and Pioglitazone ($3.56 \times 10^{-3}$). SIDD patients were also more frequently prescribed sulfonylureas ($p = 5.34 \times 10^{-4}$) which increase insulin release, and Sitagliptin ($p = 6.02 \times 10^{-3}$), a DPP-4 inhibitor which increases glucose-dependent insulin release. Patients in the SIRD cluster most frequently took anti-depressant medications ($p = 5.63 \times 10^{-3}$), medication for diabetic kidney disease e.g. Losartan ($p = 2.60 \times 10^{-2}$), and medication for rheumatoid arthritis ($4.52 \times 10^{-2}$). In the MOD cluster, medications used to treat high blood pressure and heart failure, such as Lisinopril ($8.71 \times 10^{-4}$) and esomeprazole ($p = 3.31 \times 10^{-3}$) were used. The MARD cluster also had the lowest percentage of individuals on insulin treatment ($p = 2.62 \times 10^{-4}$), whilst perindopril usage was more frequent ($4.82 \times 10^{-2}$).

## Discussion

The T2D subtype classification scheme proposed by Ahlqvist et al.[8] has been replicated in many populations[14–17,19–22,71] (Supplementary Fig. 9), but it may not be generalized to all as shown in an Asian Indian population[21]. T2D has a huge prevalence in the Middle East and North Africa region, with some of the highest rates and predicted increases over the next decade, especially in Qatar (IDF Diabetes Atlas 2021[72]). No previous study has examined the generalizability of the diabetes cluster classification scheme to Arab populations or characterized it using the latest high-throughput proteomics and metabolomics platforms. All protein and metabolite associations with T2D subtypes are made available as a resource in the Figures and Supplementary Data.

Our study shows that the T2D subtypes identified in the Scandinavian population are present in individuals of Arab/Middle Eastern descent. However, the age of diabetes onset in the QBB population was lower compared to ANDIS, especially in the MOD cluster, which could be attributed to the high incidence of obesity in younger Arab individuals. Leptin (LEP) and Follistatin-related protein 3 (FLSTL3) were significantly higher in the MOD cluster and of course increased leptin levels in obesity reflects resistance to leptin action. Also, increased

FLSTL3 levels are associated with insulin resistance and have been shown to regulate body composition and glucose homeostasis in human population studies[73,74]. Interestingly, an FLSTL3 knockout mouse has been shown to improve glucose metabolism and increase beta-cell mass[74].

Our cluster sizes are comparable between QBB and ANDIS for the most part. The difference in the SIRD group size could be due to the younger age of diabetes onset in the QBB cohort, which reflects upon the centers of clusters and ultimately the distance of samples from cluster centers. The mean age in the SIRD cluster in ANDIS is between 50–75 compared to between 30–40 in QBB. Another difference between the studies, is that most patients with T2D in ANDIS were above 50 years old at the time of diagnosis, while most patients with T2D in QBB were between 30–50 years of age at diagnosis. However, the BMI trends between ANDIS and QBB were similar, irrespective of the younger age of the QBB patients.

Among the diabetes subtypes, the MARD subtype appeared to be the healthiest group and their HbA$_{1c}$, BMI, HOMA2-B, and HOMA-IR were closest to the control group. In contrast, proteins and metabolites that were specific to SIDD were often most similar to those found in the autoimmune (SAID) group. Plexin-B2 (PLXNB2) was the most differentially regulated protein in T2D and this association was further validated in the AGES study[23]. Previous Mendelian randomization studies have suggested that PLXNB2 may have a causal effect on the development of T2D[23]. In the present study, PLXNB2 was lowest in the relatively healthier MARD subtype and closest to levels observed in controls. This could make PLXNB2 a potential drug target in individuals with subtypes other than MARD.

Cluster membership has been shown to be associated with individuals who may be more or less prone to the development of long-term complications[15,21] such as nephropathy[8,15,20] and fatty liver in SIRD[8,15], and neuropathy[20,22] and retinopathy[8,20] in SIDD. Alterations in protein biomarkers of retinopathy such as antileukoproteinase have been previously reported in the SIDD cluster[75]. We have observed an association between complement factors like Complement C2 (C2), immune-regulatory proteins like C-C motif chemokine 23 (CCL23), and other proteins of the immune system (CD59 glycoprotein (CD59), CD27 antigen (CD27), and Inhibin beta A chain (INHBA), with the SIDD subtype, suggesting that these proteins may play an important feedback role in the resolution of inflammation. A recent review highlighted a paradigm where targeting complement factors may be a possible therapeutic avenue in slowing down diabetic complications[42]. Chemokines also link obesity to inflammation and the subsequent development of insulin resistance[76]. The effects of immunomodulatory proteins on the immune system can be associated with different chronic inflammatory diseases such as diabetes, obesity, cardiovascular diseases, and cancer. For example, elevations in C2 indicate upregulation of the complement pathway in T2D, which could be a response to hypoglycemia. Hypoglycemia induces pro-inflammatory proteins such as interleukins[77]. This upregulation of the complement pathway suggests that both intrinsic and alternative pathway activation may be driven by the underlying inflammation in T2D. An inflammatory response can be rapidly induced in response to hypoglycemia and would likely drive other acute response pathways, such as the complement cascade, with significant crosstalk between the two[78]. Interestingly, 72 kDa type IV collagenase (MMP-2) was significantly lower in the SIDD subtype. MMP-2 activity has been shown to be lower in rat mesangial cells cultured in high glucose and is believed to contribute to matrix accumulation leading to the development of diabetic nephropathy[79]. Therefore, MMP-2 could potentially be developed into a protein biomarker for nephropathy in SIDD. Plexins are also receptors for semaphorins, a large family of proteins involved in various physiological processes[80]. Semaphorins are involved in a number of diabetic complications including diabetic retinopathy, nephropathy, neuropathy, osteoporosis, and wound healing[81].

Interestingly, we observed significantly higher levels of CD72 in the SIDD subtype and as CD72 appears to mediate the function of semaphorins in some immune cells[82], it may provide a potential functional link to PLXNB2. Cathepsin D (CTSD) was significantly lower in the MARD subtype and is an aspartic endopeptidase implicated in cell growth, apoptosis, and collagen biosynthesis in wounded skin of rats with diabetes and has been correlated with retinopathy and foot ulcers[17], suggesting that MARD individuals are less likely to develop such complications compared to other subtypes. CTSD also correlates significantly with HOMA-IR and the Tei index, a measure of myocardial performance[83] and may paradoxically constitute a marker for cardiac dysfunction in the more severe subtypes. However, a biomarker does not implicate causality and there may be a bidirectional relationship that requires further analysis.

Metabolic associations observed with T2D were consistent in magnitude and directionality to previously reported T2D and pre-diabetes associations[84]. Although elevated blood glucose levels are the defining feature of diabetes mellitus, multiple biochemicals alter the metabolism of fats and amino acids and are associated with impaired insulin action, obesity, and BCAA catabolic enzymatic activity[85]. BCAAs have been consistently linked to T2D development[86] and we observed alterations in BCAAs and major glucogenic amino acids. Previous studies have shown that cortisol and its metabolite cortisone, were both higher in individuals with diabetes compared to controls and altered cortisol metabolism is specifically characteristic of T2D patients requiring insulin[87]. Consistent with this observation, we observed higher cortisol in the SIDD cluster. Many amino acids have been associated with insulin resistance and decreased insulin secretion, including phenylalanine[88] and here, in SIRD, we observed significantly higher phenylalanine and 1-carboxyethylphenylalanine. Previous studies have shown that certain lipokines can affect glucose metabolism in adipose, liver, and skeletal muscle tissue[89]. For instance, a lipokine, 12,13-diHOME increases fatty acid uptake and oxidation[49] and here we show significantly higher 12,13-diHOME in the SIRD cluster. Furthermore, a significant association has been shown between 7-HOCA and asparagine and body mass index[90]. We also observed significantly higher levels of 7-HOCA and hydroxyasparagine in MOD.

The majority of proteins and metabolites which were subtype-specific were also discriminative in previous protein[23] and metabolite[29] T2D case-control studies. However, some molecules only distinguished specific subtypes but not cases from controls. This was especially true for the MARD subtype which showed levels similar to the controls. Inhibin beta A chain (INHBA), SPARC-like protein-1 (SPARCL1), and Fibronectin Fragment 4 (FN1) were significantly lower in SIDD compared to other subtypes but were not significantly different between T2D cases and controls. Also, proteins including Leptin (LEP), Phospholipase A2; membrane-associated (PLA2G2A), Follistatin-related protein 3 (FSTL3), and EGF-containing fibulin-like extracellular matrix protein-1 (EFEMP1) were significantly higher in the MOD subtype as they all associate strongly with obesity but did not differentiate between T2D cases and controls when adjusted for BMI.

Studies showing differences in protein biomarkers amongst T2D clusters are limited with the Chinese REACTION study showing that the Angiopoietin-related protein 8 (ANGPTL8) levels were significantly higher in the MARD, SIRD, and SIDD clusters compared to the MOD cluster[91]. We have found a similar trend for both ANGPTL4 and the cell-surface receptor for ANGPTL4 (TEK), two proteins that interact closely with ANGPTL8[92]. The GDS study[16], examined 77 protein biomarkers from the Olink inflammation panel and reported lower levels of Protein S100-A12 (EN-RAGE) and IL6 in SIDD compared to the other subtypes. Similarly, we also observed lower mean EN-RAGE, Interleukin-6 (IL6), and Interleukin-6 receptor subunit alpha (IL6R) levels in SIDD.

The efficacy of different diabetes drugs would be expected to differ between different subtypes according to the underlying

pathophysiology. Hence, we assessed whether ongoing treatment may reflect the clustering of individuals with T2D. Dennis et al.[14], reported differences in glycemic response among clusters in ADOPT and suggested a benefit in using thiazolidinediones for the SIRD individuals and sulfonylureas for the MARD individuals. However, in this cohort we found that sulfonylurea usage was highest in the SIDD and MOD clusters (Supplementary Data 18) whilst thiazolidinedione utilization was infrequent, reflecting reduced prescription of this class in clinical practice, and being paradoxically higher in the SIDD cluster. A current ongoing phase 2 clinical trial is investigating whether the effect of Semaglutide and Dapagliflozin differ between SIDD and SIRD individuals (ClinicalTrials.gov number, NCT04451837). The outcome of such trials may provide insight into tailored treatment plans and pave the way for personalized medicine in specific subgroups.

By identifying pathways involved in the development and progression of T2D complications, these results can be taken forward by carrying out Mendelian randomization studies of the association of cluster-specific metabolites and proteins with diabetes. This type of analysis has the potential to distinguish between potentially actionable therapeutic targets from those that are downstream of the disease and that could therefore serve as diagnostic biomarkers. This is a preliminary step to carrying out randomized control trials for drug testing. Our associations can also be used to generate hypotheses for follow-up studies on the processes that might lead to these associations.

We acknowledge some limitations of our current study. The cohort size was relatively small compared to other population-based studies and phenotyping of diabetes complications was limited to population-study-level questionnaires and biochemical measurements. The deep molecular phenotyping using the SOMAscan and Metabolon technologies provided only relative abundances of protein and metabolite levels respectively, not absolute concentrations. However, this is not a concern as the association statistics we use, i.e., linear models and t-tests, are invariant under scaling and translation of the data. The subtype-specific proteins reported here are limited to the specific protein set targeted by the SOMAscan panel, and to protein associations that are detectable in blood. Therefore, the list of subtype-specific proteins we report here is not comprehensive, and future studies using other technologies and other biological sample types may reveal further associations. Concordance and specificity of the Olink and Somalogic platforms is also an ongoing matter of discussion[93]. Some of the proteins that were not replicated on the Olink platform (ANDIS) could be a result of differences in aptamer and antibody binding. Another limitation is that since GADA was not measured, the SIDD subtype could include some individuals with autoimmune diabetes and may explain the observation of autoimmune features in this group.

In addition, outliers observed for some of the untargeted metabolomics data in the control group such as 1,5−AG, 2−hydroxybutyrate/2−hydroxyisobutyrate, and methylsuccinoylcarnitine etc. may be a result of the observed discrepancy between self-reported medication and non-targeted metabolomics as described in our previous study[94]. However, the effect of such outliers in the control group does not impact the statistical analyses amongst the subtypes, where the controls are not used. Furthermore, in the case-control analyses, numbers are large enough so that the effect of isolated outliers can be neglected.

In summary, we have identified a wealth of diabetes subtype-specific metabolite and protein signatures which have the potential to identify pathways involved in the development and progression of T2D complications, improve risk prediction, and enable more personalized treatment approaches. Our study adds further support to the medical relevance and clinical applicability of the Ahlvist et al. diabetes subtyping approach.

## Methods

### Study population

QBB includes a population of Qatar nationals or long-term residents (≥15 years living in Qatar), aged 18 years and older in the State of Qatar[95]. Extensive baseline socio-demographic data, clinical and behavioral phenotypic data, and serum concentrations of HbA$_{1c}$, triglycerides, glucose, C-peptide, creatinine, total cholesterol, LDL-C, and HDL-C, and multiple other clinical biochemistry parameters[96] have been measured at the central laboratory of Hamad Medical Corporation (HMC), accredited by the College of American Pathologists.

All QBB participants signed an informed consent form prior to their participation. The study was approved by HMC ethics committee and the QBB institutional review board under reference Ex -2019-RES-ACC-0160-0083. No compensation was given to the participants. Data collection by the Qatar Biobank was done using MS SQL Server 2008 R2. At the time of analysis, QBB data was available for 6218 participants. Over 96% of the participants reported having grandparents that were Qatari nationals. Samples selected for metabolomics and proteomics measurements correspond to the first ~3000 participants of the Qatar Biobank. No specific selection criterion was applied. 429 participants with incomplete records and 894 individuals with HbA$_{1c}$ ranging between 5.7 and 6.4 who did not match our diabetes definition (see below) were excluded, leaving 4895 samples for analysis. Blood samples were collected more than 2 h after their last meal or calorie-containing drink in 77% of participants. 50.7% of the participants (52.8% of the T2D cases and 40.6% of the controls) had been fasting for over 8 h. This dataset was split into two groups, using the samples without omics data as a training set for the clustering ($N = 2740$), and the samples with omics data as a testing set for validation, and to further evaluate the associations of the metabolite and protein levels with T2D in a case-control setting and with T2D subtypes ($N = 2155$).

The group without omics data included 631 individuals with T2D and were used to define the cluster coordinates. The group with omics data included 420 individuals with T2D and was used for cluster validation, and then further for metabolomics and proteomics associations analyses (Fig. 1). The study demographics for the two groups are shown in Table 1. Both groups of data were similar, ie. clinical variables had comparable mean values and percentages in both the T2D cases and controls (Supplementary Data 19).

### Definition of T2D and controls

Subjects were defined as controls if all the following four conditions were satisfied: first, no self-reported physician diagnosis of diabetes; second, no self-reported treatment with any diabetes-specific medication; third, HbA$_{1c}$ < 5.7%; and fourth, random glucose level <200 mg/dL. T2D was defined if any one of the following four conditions was satisfied: first, having a physician diagnosis of diabetes based on the questionnaire (13.3% of all participants), second, being treated for diabetes based on the QBB questionnaire (11.6%), third, having an HbA1c > 6.5% (10.5%), or fourth, having random glucose >200 mg/dL (3.3%). Based on this definition, 15.4% of individuals were defined as having T2D. Most individuals with a physician diagnosis of diabetes were on oral anti-diabetic treatment (73.7%), insulin treatment (23.9%), diet treatment (38.3%), and/or physical activity treatment (16.2%) (see Supplementary Fig. 10 for a Venn diagram). Individuals with HbA$_{1c}$ between 5.7% and 6.4% ($N = 894$) or self-reported gestational diabetes ($N = 4$) were excluded. Ahlqvist et al. used glutamic acid decarboxylase antibodies (GADA) to define an additional subtype of SAID. As GADA measurements were not available in QBB, individuals with self-reported type 1 diabetes (T1D) or C-peptide concentrations below 0.5 nmol/L and on insulin treatment were classified as SAID ($N = 109$). These individuals were excluded from the statistical analysis, but the proteomic and metabolic levels of this subgroup are shown where appropriate.

## Table 1 | Demographics of the QBB diabetes and control groups

| Trait | Diabetes | Controls | p value |
|---|---|---|---|
| Sample size (N = 4895) | 1051 | 3844 | — |
| Sex (male) | 444 (42.2%) | 1635 (34.5%) | 0.895 |
| Age (years) | 51.2 (11.3) | 34.8 (10.4) | $5.96 \times 10^{-261}$ |
| BMI (kg/m$^2$) | 32.2 (5.9) | 28.2 (5.8) | $7.66 \times 10^{-78}$ |
| Systolic BP (mmHg) | 125.3 (16.5) | 110.9 (13.1) | $2.28 \times 10^{-101}$ |
| Diastolic BP (mmHg) | 71.2.3 (11.3) | 67.4 (10.2) | $2.63 \times 10^{-18}$ |
| HbA$_{1c}$ (%) | 7.5 (1.8) | 5.2 (0.3) | $2.47 \times 10^{-223}$ |
| Triglycerides (mmol/L) | 1.7 (1.1) | 1.1 (0.7) | $2.89 \times 10^{-54}$ |
| LDL-C (mmol/L) | 2.9 (1.0) | 2.9 (0.9) | $2.32 \times 10^{-2}$ |
| HDL-C (mmol/L) | 1.3 (0.3) | 1.4 (0.4) | $7.68 \times 10^{-33}$ |
| Total cholesterol (mmol/L) | 4.9 (1.0) | 4.9 (0.9) | 0.463 |
| HOMA2-IR * | 2.1 (1.9) | 1.2 (0.7) | $3.47 \times 10^{-23}$ |
| HOMA2-IR (all data) | 2.8 (3.1) | 1.6 (1.6) | $1.06 \times 10^{-34}$ |
| HOMA2-B (%) * | 66.6 (54.1) | 104.9 (35.9) | $1.15 \times 10^{-46}$ |
| HOMA2-B (%) (all data) | 76.4 (64.5) | 120.9 (62.4) | $4.27 \times 10^{-77}$ |
| CPEP (nmol/L) | 1.0 (0.6) | 0.8 (0.5) | $2.04 \times 10^{-30}$ |
| Random Glucose (mg/dL) | 161.1 (72.3) | 90.2 (10.3) | $1.59 \times 10^{-155}$ |
| Fasting Glucose (mg/dL) * | 154.1 (62.7) | 90.8 (8.7) | $2.91 \times 10^{-86}$ |
| Fasting Insulin (pmol/L) * | 120.7 (270.1) | 65.8 (41.4) | $2.75 \times 10^{-6}$ |
| Insulin (pmol/L) (all data) | 146.9 (266.1) | 86.0 (100.7) | $1.08 \times 10^{-12}$ |
| Fasting time >8 h | 555 (52.8%) | 1925 (40.6%) | 0.095 |
| Family history of diabetes | 796 (75.7%) | 2491 (64.8%) | $1.86 \times 10^{-10}$ |
| Creatinine (µmol/l) | 63.2 (18.3) | 67.6 (34.4) | $2.83 \times 10^{-7}$ |
| eGFR (mL/min/1.73 m$^2$) | 99.4 (17.7) | 112.9 (14.1) | $1.29 \times 10^{-98}$ |
| Smoking | | | |
| MQ: Current smoker (cigarettes, cigar, pipe) | 107 (10.2%) | 687 (14.5%) | $2.69 \times 10^{-9}$ |
| MQ: Current smoker (water pipe) | 65 (15.5%) | 477 (21.1%) | $2.99 \times 10^{-2}$ |
| MQ: Second-hand smoker | 273 (26.0%) | 1274 (26.9%) | $1.21 \times 10^{-5}$ |
| MQ: Second-hand water pipe smoker | 191 (18.2%) | 1079 (22.8%) | $3.86 \times 10^{-11}$ |
| Physical activity | | | |
| MQ: Exercise – (>3 hr/week moderate or >1 hr/week heavy) | 82 (7.8%) | 805 (17.0%) | $1.76 \times 10^{-22}$ |
| MQ: Exercise – (>0 hr/week moderate or >0 hr/week heavy) | 136 (12.9%) | 1177 (24.8%) | $3.15 \times 10^{-30}$ |

*These clinical traits were computed for individuals who fasted for eight or more hours at the time of blood drawing.

HOMA2-IR: homeostasis model of assessment of insulin resistance. HOMA2-B: homeostasis model assessment of beta-cell function. Family history is defined as either parent having a history record of diabetes. MQ represents a question from the "main questionnaire". The second physical activity question is cumulative and includes the individuals from the first physical activity question. The data are number (%) or means (SD), as appropriate, p values are from Fisher or two-sided student t tests for categorical or continuous variables, respectively.

### Training and testing sets

The cohort was split into two sets, a training set (N = 2740) and a testing set (N = 2155). The latter was chosen to overlap with available proteomics, metabolomics, and medication usage data. There were no substantial differences in demographics between the training and the testing set (Supplementary Data 19). The training set included 631 individuals with T2D, and the testing set included 420 individuals with T2D.

### Medication

QBB study participants provided information on their regular usage of over-the-counter and prescription medication as free text, which required annotation and homogenization. The questionnaire included the following question: "Are you taking any over-the-counter medication or prescription medicines regularly? For example, daily, weekly, monthly, or every few months - such as depot injections?" and allowed participants to provide up to 30 free text entries. Anatomical Therapeutic Chemical (ATC)[97] annotation was retrieved from the DrugBank annotation file. We annotated all entries from the questionnaire with a unique active molecule from the DrugBank repository, molecular class, indication, and its corresponding ATC code where available. The medication data covered 394 unique molecules, 529 ATC codes, 218 molecular classes, and 117 indications.

### Proteomics

Levels of 1305 blood circulating proteins (Supplementary Data 20) were measured for 2935 samples using the aptamer-based SOMAscan platform (kit version 1.3, Somalogic, Boulder, CO)[98] implemented at Weill Cornell Medicine – Qatar, as previously described[99]. A detailed description of the platform can be found in the "SOMAmer Reagent Specificity Technical White Paper SM-500–102015", which was originally available on Somalogic's website http://info.somalogic.com/hubfs/January_2016/SSM-002-Rev-3-SOMAscan-Technical-White-Paper.pdf (accessed November 26, 2016) and is now archived and available at https://studyres.com/doc/7837606/technical-white-paper. Briefly, EDTA-plasma was incubated with bead-coupled epitope-specific aptamers (SOMAmers). Bead-bound proteins were then biotinylated and complexes comprising biotinylated target proteins and fluorescence-labeled SOMAmers were photocleaved and recaptured on streptavidin beads. SOMAmers were then eluted and quantified by hybridization to custom arrays of SOMAmer-complementary oligonucleotides. Data was used as provided by Somalogic and includes their in-house batch normalization steps. The resulting raw intensities were processed using different standards as a reference, including hybridization normalization, median signal normalization, and signal calibration to control for inter-plate differences. No samples or data points were excluded. Overlapping phenotype data were obtained for 2155 of the 2935 samples and proteomics data for these samples were used. Quality control was performed using repeated measures of two QC samples. The median coefficient of variance (CV) was 7.3% for both QC samples, based on 51 and 54 repeated measures, respectively. 95% of the aptamers had a CV below 17.2% and 17.6%, respectively, and 5% had a CV below 4.6% and 4.1%, respectively.

### Metabolomics

1159 metabolites (937 named compounds and 222 compounds of unknown structural identity) were quantified using Metabolon HD4 technology (Metabolon Inc., Durham, NC) (Supplementary Data 21) for 3000 samples as previously described[100,101]. All measurements were performed on a Metabolon HD4 platform implemented at the Anti-Doping Laboratory in Qatar (ADLQ) under a joint laboratory agreement with Metabolon and support from Weill Cornell Medicine – Qatar, the Qatar Biomedical Research Institute, and the interim Translational Research Institute (iTRI) of HMC[94]. For 2155 of the 3000 samples, we obtained overlapping phenotype data for this study. Metabolomics data for these samples were used. Data was used as provided by Metabolon and includes their in-house batch normalization steps. No additional QC was applied that led to exclusion of samples. Metabolites with more than 90% missingness were excluded. Instrument variability, based on measurement of internal standards,

was 12% and total process variability, based on endogenous bio-chemicals measured in repeated reference samples, was 16%.

## Statistical analysis

Statistical analysis was conducted using R (version 4.0.5) and RStudio (version 1.4.1106). *T* tests, Fisher exact tests, and linear and logistic regression models with covariates as indicated were conducted as appropriate based on the respective variable types, using subroutines implemented in base R. Multiple testing was accounted for using conservative Bonferroni correction at a significance level of 0.05 divided by the number of tests conducted in the specific cases (number of metabolites, proteins, traits taken forward for replication, etc. as indicated).

## Cluster analysis

Model parameters were selected based on five commonly measured variables as in Ahlqvist et al.[8]. We used BMI, age at onset of diabetes, and homeostasis model assessment HOMA2 estimates of ß-cell function (HOMA2-B) and insulin resistance (HOMA2-IR) based on C-peptide concentrations calculated with the HOMA calculator (University of Oxford, Oxford, UK)[102]. Patients defined as SAID (see above) were excluded from the clustering and assigned to their own subtype, and clustering was carried out on the patients with T2D only. Cluster analysis was carried out on standardized values centered to mean = 0 and s.d. = 1. The optimal number of clusters was determined from the training set using the Mclust function in the "mclust" R library v.5.4.5. Clusters coordinates were identified in the training set of QBB individuals ($N = 631$) and applied to a separate QBB set which was deeply phenotyped ($N = 420$). We determined the optimal number of clusters using the Bayesian Information Criterion (BIC) for expectation-maximization, initialized by hierarchical clustering for parameterized Gaussian mixture models. We computed the BIC for various cluster sizes (two to 15). Using *k*-means cluster analysis on standardized variables, we derived cluster center coordinates for different values of *k* (Supplementary Fig. 11). Using a voting scheme[103], we determined the optimal number of clusters to be $k = 4$ (Supplementary Fig. 12). This finding was consistent with that observed by Ahlqvist et al. Cluster stability was assessed by Jaccard similarity[104] using 2000 re-runs of the *k*-means procedure. The Jaccard similarity to the original clusters was greater than 0.75, which is generally considered an acceptable threshold for cluster stability[104]. The cluster variables in QBB followed a similar trend to the Swedish All New Diabetics in Scania cohort (ANDIS). We, therefore, assigned cluster labels based on the five clinical variable averages that were characteristic of each T2D subtype following Ahlqvist et al.

## Robustness and reproducibility analysis

To ensure robustness and reproducibility of our results, we undertook a number of sensitivity tests on the cluster analysis. First, we replicated the clustering in the testing set and compared that to using cluster coordinates from the training set. In the case of clustering in the testing set, the assigned clusters from the *k*-means algorithm were used directly. However, when using the training set coordinates, cluster membership was determined by assigning every individual in the testing set to the cluster with the minimum Euclidean distance to the training coordinates. We then repeated the clustering with varying numbers of clusters. We also repeated the cluster analysis separately for females and males. As QBB participants were not all in a fully fasting state, we further tested the sensitivity of the cluster assignments to self-reported time-since-last-meal. Finally, we tested whether ANDIS cluster centers could be used directly to classify QBB participants into T2D subtype clusters.

## Identification of T2D-associated proteins and metabolites

For proteins, logistic regression models (glm) were used to test for association with the T2D state, including age, sex, and technical covariates into the model. Technical covariates include log(HSP90) which is a measure of cell lysis, week of sample collection, fasting minutes, and SomaLogic tube number. For metabolites, linear regression models (lm) were used to test for association with T2D using age, sex, BMI, and technical covariates into the model. Different types of models (glm for proteins and lm for metabolites) were used for consistency with previously published work[23,29]. Protein and metabolite levels were log-scaled, *z* scored, and outliers were winsorized to 5 s.d. before computing the association.

## Replication of association of T2D with proteomics

Association data for the replication of the T2D protein association was obtained from the published AGES-Reykjavik study[23]. The AGES-Reykjavik study was approved by the National Bioethics Committee in Iceland (approval number VSN-00-063), the National Institute on Aging Intramural Institutional Review Board (U.S.), and the Data Protection Authority in Iceland. All participants provided written informed consent. In that study, serum levels of 4137 proteins, targeted by 4782 SOMAmers, were measured at SomaLogic (Boulder, CO) in samples from 5457 AGES-Reykjavik participants, as previously described[105]. The AGES-Reykjavik cohort included 654 individuals with T2D and 4784 controls. After applying a Box-Cox transformation on the protein data, associations between serum protein levels and prevalent or incident T2D were determined using a logistic regression adjusted for age and sex. After following the same preprocessing steps and statistical methods, we replicated the associations for proteins that were shared between QBB and AGES ($N = 107$).

## Replication of association of T2D with metabolomics

Metabolomics associations with Type 2 Diabetes (T2D) have been previously reported for the QMDiab study using the Metabolon HD2 platform[29,106]. However, here we are using data that has been recently remeasured on the more recent Metabolon HD4 platform in Durham (NC), which is compatible with the QBB metabolomics data annotation. The QMDiab was approved by the Institutional Review Boards of HMC and Weill Cornell Medicine, Qatar (WCM-Q) under research protocol #11131/11. Written informed consent was obtained from all participants. Data for 1104 metabolites from 309 samples of QMDiab were used. From 194 metabolites that associated with T2D in QBB at a Bonferroni level ($p < 0.05/1159 = 4.31 \times 10^{-5}$), data was also available for 175 metabolites in QMDiab. For the replication, the same processing of the metabolomics data was performed, including log-scaling, *z* scoring, and outlier winsorization to 5 s.d., before computing the association.

## T2D subtype cluster omics and medication analysis

Cluster-specific proteins and metabolites were identified using linear models without covariates and following two criteria. First, the omics levels for a given cluster were compared to all others combined, requiring Bonferroni significance levels ($p < 0.05/N$metabolites and $p < 0.05/N$proteins). Second, the omics levels for a given cluster were compared to all other clusters individually, requiring nominal significance ($p < 0.05$). Cluster-specific drug usage was identified using a Fisher test comparing usage of a given drug in a given cluster to all other clusters combined, requiring nominal significance ($p < 0.05$).

## ANDIS study

The ANDIS study included patients with newly diagnosed diabetes ($n = 8980$) from the Swedish All New Diabetics in Scania cohort. The ANDIS study protocol was approved by the regional ethics review committee in Lund (ANDIS:584/2006 and 2012/676). All participants provided written informed consent. This cohort was used in the original study that first defined the T2D subtypes[8]. Data-driven k-means cluster analysis using six variables (glutamate decarboxylases antibodies, age at diagnosis, BMI, HbA1c, and HOMA2-B, and HOMA2-IR,

was carried out on these patients. Five distinct clusters of diabetes subtypes with significantly different patient characteristics and risk of complications were identified. These subtypes included one auto-immune or SAID, and four subtypes of type 2 diabetes, namely SIDD, SIRD, MOD, MARD.

### Replication of associations of T2D subtypes with proteomics

Replication of the subgroup-specific proteins was attempted in the AGES-Reykjavik study[23] and the ANDIS study[8] using a linear regression model for each cluster versus all other clusters, while adjusting for sex. The protein data and T2D subgroup definition in AGES has been previously described[23]. Protein levels were measured in the ANDIS study for $N = 176$ individuals (44 individuals per subtype) using the Olink platform (Olink Proteomics, Uppsala, Sweden). Equal numbers of men and women were selected based on Euclidean distance to the cluster centers to be representative of their subtype. All selected individuals were between age 38.1 and 75.2 years, of European decent, and had their blood samples taken within 3 months of diabetes diagnosis. The following 13 Olink panels were used: Olink CARDIOMETABOLIC, Olink CARDIOVASCULAR II, Olink CARDIOVASCULAR III, Olink CELL REGULATION, Olink DEVELOPMENT, Olink IMMUNE RESPONSE, Olink INFLAMMATION, Olink METABOLISM, Olink NEURO EXPLORATORY, Olink NEUROLOGY, Olink ONCOLOGY II, Olink ONCOLOGY III, and Olink ORGAN DAMAGE, covering a total of 1161 distinct protein assays. The biomarker expression was measured using logarithm of the relative biomarker/protein concentration in each panel, expressed as normalized protein expression values. Outlier analysis was performed using an unsupervised clustering algorithm using a One Class Support Vector Machine. Four samples, one from each subtype, were identified as outliers and excluded from the analysis, leaving $N = 172$ individuals (43 individuals per subtype). Data was available for 30 of the 47 subtype-specific proteins (matched by Uniprot identifiers) that were shared between the Olink data in ANDIS and the Somalogic data in QBB.

### Reporting summary

Further information on research design is available in the Nature Portfolio Reporting Summary linked to this article.

## Data availability

The QBB data are available under restricted access for the informed consent given by the study participants does not cover posting of participant-level phenotype data in public databases, access can be obtained in the form of an MS SQL Server 2008 R2 database, upon request from QBB (https://www.qatarbiobank.org.qa/research/how-to-apply). Requests are submitted online and are subject to approval by the QBB board. The custom-design Novartis SOMAscan is available through a collaboration agreement with the Novartis Institutes for BioMedical Research (lori.jennings@novartis.com). Data from the AGES Reykjavik study are available through collaboration (AGES_data_request@hjarta.is) under a data usage agreement with the Icelandic Heart Association. The QMDiab data are available under restricted access for the informed consent given by the study participants does not cover posting of participant-level phenotype data in public databases. Access can be obtained in the form of an R data file, upon request from the corresponding author. Data from the ANDIS study are available upon request from the ANDIS steering committee (emma.ahlqvist@med.lu.se). All data supporting the findings described in this manuscript are available in the article and in the Supplementary Information and from the corresponding author.

## Code availability

Standard statistical analysis was carried out using functions implemented in R Studio version 4.0.5 and version 1.4.1106. Documentation on how to use the standard built-in R functions can be found at https://www.rstudio.com/products/rpackages/.

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

## Acknowledgements

We are grateful to all study participants of Qatar Biobank for their invaluable contributions to this study. We thank Olle Melander for his suggestions and comments on the present manuscript. This study is supported by the Biomedical Research Program at Weill Cornell Medicine in Qatar, a program funded by the Qatar Foundation, and by QNRF grant NPRP11C-0115-180010. Qatar Biobank is supported by Qatar Foundation. The statements made herein are solely the responsibility of the authors. EA was funded by grants from the Swedish Research Council (2020-02191) and the Novo Nordisk foundation (NNF18OC0034408). Olink measurements in ANDIS were sponsored by Olink Proteomics (Uppsala, Sweden). The AGES-Reykjavik study was funded by the Icelandic Heart Association contract HHSN271201200022C, National Institute on Aging contract N01-AG-12100, and Althingi (the Icelandic Parliament). V.G. is supported by the University of Iceland postdoctoral fund and the Icelandic Research Fund (206692-051). The protein measurements in AGES were supported by the Novartis Institute for Biomedical Research and performed at SomaLogic.

## Author contributions

Conceived and designed the study: S.Z., K.S. Analyzed data: S.Z. Contributed reagents/materials/analysis tools: Va.G, N.S., M.T. Wrote the paper: S.Z., A.H, Va.G., Vi.G., L.J., E.A., R.M., O.A., A.B.A., K.S. All authors discussed the results and reviewed the final manuscript.

## Competing interests

L.L.J. is an employee and stockholder of Novartis. The remaining authors declare no competing interests.
