## [Peer Review File · Nature Communications]

Metabolic and proteomic signatures of type 2 diabetes subtypes in an Arab populationReviewers' comments:

Reviewer #1 (Remarks to the Author):

1. Abstract. The use of references in the abstract does not facilitate understanding of messages. Furthermore, the abstract is written in a medical style but the messages of the manuscript are rather supporting future therapies and are further mechanistic or at least epidemiological.
2. Introduction. Please differentiate the stratification into four subtypes of diabetes type 2 to that proposed by Leif C. Groop. What are the advantages and consequences of such classifications?
3. Why was the stratification according to Ahlquist was used in this study?
4. In the definition of T2D and controls. Please describe if any individuals were diagnosed post metabolomics analyses due to e.g. high hexose level or any other data.
5. Please verify if any of the outliers on the control group might have used a medication not reported in a questionnaire. This could be derived from untargeted metabolomics data by selective analyses for pain killers, statins, anti-inflammatory agents, metformin or other T2D-related drugs.
6. Were any metabolite concentration imputed? Were those used in the cluster discrimination?
7. Please demonstrate what is overall explanation of variability of metabolomics or proteomics data by clusters in unsupervised multivariate data analysis like PCA.

Reviewer #2 (Remarks to the Author):

Zaghlool et al have conducted a very interesting study looking at diabetes in a Qatari Arab population, clustering the cases based on physiological parameters and comparing how well they replicated the original findings of Ahlqvist et al. (2018). In addition, they extended the study by looking at proteomic and metabolomics differences between the clusters.

The study is interesting, well conducted, has reasonable sample sizes to start with (although some of the eventual clusters end up quite small), and overall has strong merit to be published. It is replicating the findings of at least five previous studies, based on different populations around the world, although it is the first to my knowledge to concentrate on Qatar or the Middle East. Most of the associations it mentioned have been highlighted in other studies. However, the study has merit in its own right, both due to its unique population, and because it validates previous observational cohorts. I would recommend that the study addresses more directly the underlying connections between the proteomics and metabolomics data. It appears to be analysing the datasets almost as independent of each other in the discussion and results. Given the heavy bioinformatics focus of this paper, I would have liked to have seen an attempt at merging the results from the two datasets to explain them together. If this was attempted and the results were not easy to interpret, then a paragraph or two in the paper explaining this would be worthwhile; it may help other researchers in the future. Likewise, I would have liked to have seen more analysis of confounding factors and how much they may be contributing to the final classification.

Little mention is made of effects of diet and microbiome on cluster outcomes and how this may be influencing both BMI, insulin resistance and overall tendency to develop diabetes. In addition, both are likely to affect population tendencies to develop diabetes and the metabolites later measured. I would also draw the authors' attention to an article which came out late last year on a similar theme, all-be-it in a different population. Roderick C. et al. Distinct Molecular Signatures of Clinical Clusters in People With Type 2 Diabetes: An IMI-RHAPSODY Study. *Diabetes* 1 November 2021; 70 (11): 2683–2693. <https://doi.org/10.2337/db20-1281>. They may be interested in comparing their findings.

Specific points

Line 115: I would specify physician diagnosis of diabetes mellitus to distinguish it from the much rarer diabetes insipidus (which I have assumed was excluded from this study of diabetes)

Line 165: was it random which patients had phenotype and metabolomics/proteomics data? Or was this in any way biased.

Line 168: please clarify in the text if you have given C.Vs as a percentage (0.172%) or as a decimal i.e. 17%

Line 170 onwards: Nothing was said here about any quality assurance of the metabolomics datasets.

For example, were any quality control measures applied to the metabolomics data and were any compounds excluded from further analysis due to poor technical reproducibility? How were batch differences accounted for in measurements of proteomics and metabolomics datasets, especially given the semi-quantitative nature of most of the measurements.

Line 251: Just a comment: you used Bonferroni to correct for multiple testing. This is not wrong, but it may be overly strict, especially given the inherent co-dependency of many of the metabolites.

P11 Line 285: reference is made to the cluster sizes being broadly comparable between QBB and ANDIS. The authors mention a difference in the SIRD group sizes; do they have an explanation for this? Could it be related to the younger age of onset? Or the cut off values used for the clustering?

P298: the authors mention that clustering male and females separately showed little difference in eventual cluster membership, but was there any male/female difference in cluster membership, or the average cluster values? I was particularly interested in whether individuals on the borderline of different cluster groups were more likely to be one sex, and whether menopause was a confounding factor in cluster membership. Do males and females tend to have different outcomes in their clusters?

Table 1: please clarify is the percentage in the exercise groups are cumulative or separate i.e. were individuals who exercised over 3 hours a week counted in both groups.

Time since disease was diagnosed would also be nice to include here – and if possible, an analysis to confirm that different classifications are genuinely separate diagnoses and not marking a linear progression of diabetes.

Several supplementary tables were referenced in the main paper but no supplementary tables were available in the copy I had to review.

Line 404: How certain are the authors that the plasma level of phosphate was comparable to the in-vivo measurement and not a post-collection artefact?

Line 468 onwards: the authors have mentioned specific medication differences between clusters, but were these checked for their confounding effects on the eventual results, especially since some of the medications would be immunomodulatory.

Line 518: it was not clear to me how you had made the link between association of certain immunomodulatory proteins and their role in a feedback resolving inflammation.

Reviewer #3 (Remarks to the Author):

This study extended the Ahlqvist et al. diabetes subtype findings to a large Arabic cohort by 1) (somewhat) confirming similar clinical diabetes subtypes by using both de novo clustering methods in a training dataset of 2,740 individuals that were then evaluated in the testing set of 2,155 and 2) leveraging metabolomics and proteomics to profile circulating biomarkers associated with these subtypes in the testing cohort of 2,155 individuals. The SOMAscan 1,300 plex proteomics platform and Metabolon metabolomics platforms were used. Strengths of the study include the novelty of evaluating the Ahlqvist et al. diabetes subclassification system in a new population with high T2D burden, and extending the phenotyping to metabolomic and proteomic profiling to identify circulating biomarkers associated with this novel diabetes classification system. One significant limitation was that fact that was a lot of variability in the time fasting prior to blood collection (with 23% having not fasted) and that only 50% fasted for 8 hours which can significantly affect metabolite—and to a possibly lesser extent protein concentrations. Another major concern was the small numbers considering that these metabolite and protein associations were determined based on 420 individuals with diabetes that were subdivided into 4 classes (with one subclass only containing 17 individuals).

Major (detailed) Critiques:

1. Why was BMI adjustment only completed for the metabolomics analyses and not for the proteomics analyses? The rationale provided in the manuscript was not sufficient.
2. Please clarify the exact methods used for the weighting in each cohort. This was quite unclear: “After using the ANDIS coordinates instead of the QBB coordinates to classify 920 QBB patients, 35% changes in the cluster assignments were observed.” Please explain.
3. Given the variability in fasting status prior to blood sample collection, sensitivity analyses were conducted for the diabetes subtype clustering. However, were sensitivity analyses conducted for the metabolomics and proteomics analyses? Fasting is known to influence metabolite levels, and possibly to a lesser degree, protein levels.

4. Given the technical differences between the SOMAScan and Olink platforms, what was done to align the relative aptamer/protein concentrations between QBB and ANDIS?
5. Very few metabolites validate on the Olink platform. What were the correlations between the proteins measured on both platforms if there were any samples run on both platforms? And more generally, can the authors supply a table to inform us what data support the specificity of the platform(s)?
6. I struggle to understand the significance of Table 2.
7. In Figure 6, how were these proteins and metabolites selected? What additional information does this figure provide compared to Figure 4 and 5?
8. Please clarify in the limitations section why the measurement of relative abundance of proteins and metabolites are “not a concern in this kind of study.”
9. The variability in hours fasted prior to blood draws should be discussed in the limitations given its relevance especially to the metabolomics analysis.
10. Why do the authors exclude pre-diabetes?
11. Why was the metabolomics replication so poor? “... 41 (23%) of these associations were statistically significant at a Bonferroni level ($p < 0.05/175$)”. Isn't that a major concern?
12. It is still not clear to me how much of the biochemical differences are due to the underlying med use...much less BMI or other clinical traits driving the subclasses.

Minor Critiques:

1. Please define and briefly describe the ANDIS trial for readers who are not aware this is the population that the original Ahlqvist et al. diabetes clustering was derived from.

In summary, the assignment of individuals into these various sub-classes will be particularly useful if 1) they respond differently to interventions or 2) if the metabolomics/proteomics inform causative biology. The second point is the focus of this paper, but it seems early as one reads through the discussion. The associations seem to just be downstream of the clinical features that distinguish the groups (age, BMI, etc...)

Reviewer #4 (Remarks to the Author):

Zaghloul et al. show that previously established type 2 diabetes subtypes derived from Swedish data replicate in an Arab population. These subtypes have been replicated in various other populations as well, but now for the first time also in Arabs. Further, the authors analysed metabolic and proteomic data in relation to these subtypes and provide insights into the underlying etiology of these subtypes. While the study is well-conducted and the findings are supported by partial replication in other studies and literature, I would like to see more discussion and better presentation of the novel aspect of this study regarding the metabolic and proteomic signatures. Below I detail my comments.

1. It seems that replication of T2D-associated proteins and metabolites was attempted in AGES and QMDIAB but cluster-specific proteins were replicated in ANDIS. Is there a reason why the clustering and cluster-specific associations were not attempted in AGES and QMDIAB?
2. Validation of clusters. Please specify the parameters/tests that were used to assess the fit of the clusters in the test set.
3. Why was a different transformation for proteins (Box-Cox vs. log) used in the replication data AGES? Did the authors check whether this would influence the results.
4. The clusters were transferable to the Arab population but with population-specific coordinates (Figure 2B and C). I think this should be highlighted better in the text (and abstract), especially regarding potential clinical utilisation of the clusters.

5. How would the authors take the results further? In the conclusions it is stated that the findings “have the potential to identify novel pathways involved in the development and progression of T2D complications, improve risk prediction, and enable more personalized treatment approaches”. I would hope to see a more detailed discussion and perhaps some specific figures on this aspect, which seems to me to be the most interesting part of the paper.

Minor comments:

1. Abstract. Currently it is unclear by only reading the abstract that the n relates to the number of metabolites/proteins rather than individuals.
2. Table 1 smoking: is shisha in second-hand smoking the same as water pipe in the current smoker?
3. P.8 line 207: define ANDIS.
4. P. 12 line 323: replicate “in” other populations
5. P. 19 line 483: define MENA.
6. P. 19 line 501: should it be HOMA2-IR?

Point-by-point response to the reviewers' comments

Reviewer #1 (Remarks to the Author):

1. Abstract. The use of references in the abstract does not facilitate understanding of messages. Furthermore, the **abstract is written in a medical style** but the messages of the manuscript are rather supporting future therapies and are further mechanistic or at least epidemiological.

Response:

Thank you, we have removed the references from the abstract and revised the abstract in a less medical and more mechanistic tone:

ABSTRACT

Background. Type 2 diabetes (T2D) has a heterogeneous etiology which may influence its progression, treatment choices and risk of complications. A data driven cluster analysis in four separate European populations of patients with T2D previously identified four subtypes: severe insulin dependent (SIDD), severe insulin resistant (SIRD), mild obesity-related (MOD), and mild age-related (MARD) diabetes. Our aim was to apply this classification to an Arab population in Qatar and to further characterize the metabolomic and proteomic signatures in these subtypes.

Methods. The subtype clustering approach was applied to 631 individuals with T2D from the Qatar Biobank (QBB) and validated in an independent set of 420 participants from the same population. Cluster-specific signatures of circulating metabolites (N=1,159) and proteins (N=1,305) were established, and cluster-specific protein signatures were tested for replication in an independent study with 588 participants.

Findings. The four subtypes of T2D were validated in the Arab cohort. Cluster-specific metabolomic and proteomic associations revealed subtype-specific molecular mechanisms, including activation of the complement system with several features of autoimmune diabetes and reduced 1,5-anhydroglucitol (1,5-AG) in SIDD, impaired insulin signaling in SIRD, and elevated leptin and fatty acid binding protein levels in MOD. Participants in the MARD cluster appeared to be the healthiest subgroup with metabolomic and proteomic profiles most similar to the control group.

Interpretation. We have translated the T2D subtype system to an Arab population and further identified distinct molecular signatures to further our understanding of the etiology of these subtypes.

2. Introduction. Please differentiate the stratification into four subtypes of diabetes type 2 to that proposed by Leif C. Groop. **What are the advantages and consequences of such classifications?**

Response:

Our stratification scheme into four out of five type 2 diabetes clusters is de facto identical to that proposed by Groop et al. [Ahqvist et al. Lancet Diabetes Endocrinology; 2018]. We differ only in the definition of the type 1 diabetes cluster, which was based on anti-GAD measurements by Groop et al. Due to the lack of anti-GAD measurements we use an alternative criterion. To clarify this, we have added the following text to the introduction:

“Diabetes is presently classified into type 1 and type 2 diabetes. Ahqvist et al. used data on age at diagnosis, BMI, HbA1c, homeostasis model assessment (HOMA) estimates of beta-cell function (HOMA2-B), insulin resistance (HOMA2-IR), and glutamic acid decarboxylase antibodies (GADA) to stratify subjects into four clusters representing T2D subtypes (SIDD, SIRD, MOD, and MARD) and one cluster with severe autoimmune diabetes (SAID), which represents type 1 diabetes. We follow the same stratification for those with type 2 diabetes, however, for those with type 1 diabetes we use C-peptide levels rather than GADA.”

3. Why was the stratification according to Ahlquist was used in this study?

Response:

We understand that the reviewer is referring to the paper by Ahlqvist et al, but using the senior author’s (Groop LC) name. Indeed, since its publication in 2018, the paper has been cited over 1,200 times and has become the reference standard for subtyping patients with diabetes in several different populations. However, to date it has not been validated in an Arab population.

4. In the definition of T2D and controls. Please describe if any individuals were diagnosed post metabolomics analyses due to e.g. high hexose level or any other data.

Response:

Thank you for requesting this clarification. The QBB cohort subjects with diabetes not identified just from history, but also from blood glucose and HbA1C; ~3% of subjects with diabetes included were newly-diagnosed. We did not identify additional subjects with diabetes from the metabolomic data. This has been clarified in the Methods:

“T2D was defined if any one of the following four conditions was met: first, having a physician diagnosis of diabetes based on the questionnaire (13.3% of all participants), second, being treated for diabetes based on the QBB questionnaire (11.6%), third, having an HbA1c > 6.5% (10.5%), or fourth, having a random glucose > 200 mg/dL (3.3%). Based on this definition, 15.4% of individuals were defined as having T2D.”

5. Please verify if any of the outliers on the control group might have used a **medication not reported in a questionnaire**. This could be **derived from untargeted metabolomics** data by selective analyses for pain killers, statins, anti-inflammatory agents, metformin or other T2D-related drugs.

Response:

We agree that some of the outliers in the control group could have T2D and could be identified using drug metabolites in their blood. We have addressed this issue in a recent publication [Suhre et al. Metabolites 2022]. Although self-reported drugs and their metabolites were detected in 79.4% of cases, only 29.5% of detected drug metabolites matched to self-reported medication. Possible explanations for differences include under-reporting of over-the-counter medications by the study participants, such as paracetamol, mis-annotation of low abundance metabolites, such as metformin, and inability of the current methods to detect them. Based on these findings we do not believe that the definition of T2D can be substantially improved using this approach. In any case, these outliers mainly concern the control group, which was not used in the statistical analysis of the subgroup of patients with T2D. We have added the following text to the caveat section of the manuscript.

“In addition, outliers observed for some of the untargeted metabolomics data in the control group such as 1,5-anhydroglucitol, 2-hydroxybutyrate/2-hydroxyisobutyrate, and methylsuccinoylcarnitine etc. may be a result of the observed discrepancy between self-reported medication and non-targeted metabolomics as described in our previous study [Suhre et al. Metabolites 2022]. However, the effect of such outliers in the control group does not impact the statistical analyses amongst the subtypes, where the controls are not used. Furthermore, in the case-control analyses, numbers are large enough so that the effect of isolated outliers can be neglected.”

6. Were any **metabolite concentration imputed**? Were those used in the cluster discrimination?

Response:

No metabolite concentrations were imputed and only samples with complete clinical parameter data sets were included in the study. Furthermore, metabolite concentrations were not used in the cluster discrimination.

7. Please demonstrate what is **overall explanation of variability of metabolomics or proteomics data by clusters in unsupervised multivariate data analysis like PCA**.

Response:

Following the reviewer's suggestion, we have carried out PCA analysis of the proteomics and metabolomics data. Some of the PCs variance can be observed in certain subtypes, such as proteomics PC2 and PC4 associating more strongly with MARD compared to the other subtypes, and proteomics PC7 and PC9 associating more strongly with SIDD. As the interpretation of PCA loadings is not intuitive, we did not pursue PCA analysis further in our study. We have added the following text to the manuscript:

"We computed the explained variance of all proteomic and metabolomic PCs, by the cluster membership, using a linear regression model. Proteomics PC2, PC7, and PC4 yielded the highest R^2 of 15.5%, 14.3%, and 10.0%, respectively. Similarly, the cluster membership explained 18.2%, 7.5%, and 7.5% of the variance in metabolomics PC3, PC7, and PC9, respectively. However, as the interpretation of PCA analysis is challenging, we did not seek further explanations in this study."

Reviewer #2 (Remarks to the Author):

Zaghloul et al have conducted a very interesting study looking at diabetes in a Qatari Arab population, clustering the cases based on physiological parameters and comparing how well they replicated the original findings of Ahlqvist et al. (2018). In addition, they extended the study by looking at proteomic and metabolomics differences between the clusters.

Response:

We thank the reviewer for their interest in our study.

The study is interesting, well conducted, has reasonable sample sizes to start with (although some of the eventual clusters end up quite small), and **overall has strong merit to be published**. It is **replicating the findings of at least five previous studies**, based on different populations around the world, although it is the **first to my knowledge to concentrate on Qatar or the Middle East**. Most of the associations it mentioned **have been highlighted in other studies**. However, the study has merit in its own right, both due to its unique population, and because it validates previous observational cohorts.

Response:

We thank the reviewer for acknowledging that this study has merit to be published as it is the first to replicate findings in an Arab population.

The reviewer states that most of the associations we report have been highlighted in other studies. We need to clarify that this is only true for our omics- diabetes case-control associations

which were only carried out as a pre-analysis step to serve as a quality measure for our data to show that on a case-control basis we observe similar signals to previous studies.

The real focus and novelty of this study is in the identification of novel proteomic and metabolomic signatures for each subtype of diabetes. At the time of submission, no previous study had reported any omics-cluster associations. In the meantime, a very recent study [Slieker et al. Diabetes 2021] has been published with a similar analysis in a European population. In the revised manuscript we have explicitly compared our results and overlap between both populations.

I would recommend that the study **addresses more directly the underlying connections between the proteomics and metabolomics data**. It appears to be analyzing the datasets almost as independent of each other in the discussion and results. Given the heavy bioinformatics focus of this paper, I would have liked to have seen **an attempt at merging the results from the two datasets** to explain them together. If this was attempted and the results were not easy to interpret, then a paragraph or two in the paper explaining this would be worthwhile; it may help other researchers in the future.

Response:

Thank you, we believe we have extensively described the cluster-specific proteomic and metabolic signatures side-by-side in an integrated manner throughout the results and discussion.

However, following the reviewer's suggestion, at an attempt to merge the result from the two datasets, we added the following text to the Results:

*"To combine the results from the metabolomics and proteomics datasets, we conducted a correlation analysis between the cluster-specific proteins and metabolites (**Supplementary Figures 9, 10, 11**). Some correlation can be observed between the proteins and metabolites, some of which are within the same subtype, particularly in the MOD subtype, while others are between different subtypes. These results are not easy to interpret due to confounding factors such as sex, age, etc. which may be driving some of these cross-subtype associations. Further analysis is beyond the scope of this study."*

Supplementary Figure 9. Heatmap of the correlations between the identified cluster-specific proteins (y-axis) and the cluster-specific metabolites (x-axis). Correlation coefficients are computed using Pearson's correlation. The proteins and metabolites are color-shaded by the relevant subtype they are associated with (SIDD, SIRD, MOD, and MARD) as used throughout the manuscript.

Supplementary Figure 10. Heatmap of the correlations across the identified cluster-specific proteins (both x-axis and y-axis). Correlation coefficients are computed using Pearson's correlation. The proteins are color-shaded by the relevant subtype they are associated with (SIDD, SIRD, MOD, and MARD) as used throughout the manuscript.

Supplementary Figure 11. Heatmap of the correlations across the identified cluster-specific metabolites (both x-axis and y-axis). Correlation coefficients are computed using Pearson's correlation. The metabolites are color-shaded by the relevant subtype they are associated with (SIDD, SIRD, MOD, and MARD) as used throughout the manuscript.

Likewise, I would have liked to have seen **more analysis of confounding factors** and how much they may be contributing to the final classification.

Response:

Following the reviewer's suggestion, we have analysed the association of the following confounding factors with the cluster assignment: fasting time, lipid lowering drugs, diabetes and

hypertension drugs, smoking, sex, time since diagnosis, and a polygenic risk score. We computed the polygenic risk score for type 2 diabetes using variants and weights from the previous study of Khera et al. [Khera, A. V. et al, Nature Genetics 2018]. Briefly, the score is based on summary statistics from a recent GWAS with type 2 diabetes and assigns weights to each genetic variant depending on the strength of its association with diabetes. The following confounders (marked in red) were significantly associated with the cluster membership – diabetes drugs, hypertension drugs, sex, and time since diagnosis.

Confounder	Association P-value
Fasting time	lm (fasting_time ~ cluster_membership) P = 0.08147
Diabetes Genetic Score	lm (diabetes_genetic_score ~ cluster_membership) P = 0.4471
Smoking	Chi-squared test P = 0.1643
Lipid lowering drugs	Chi-squared test P = 0.2956
Diabetes drugs	Chi-squared test P= 8.54E-05. 73% in SIDD 35% in SIRD 52% in MOD 45% in MARD
Hypertension drugs	Chi-squared test P = 0.0043. 31% in SIDD 29% in SIRD 46% in MOD 24% in MARD
Sex	Chi-squared test P=5.28E-06 51 males and 42 females in SIDD 11 males and 6 females in SIRD 23 males and 66 females in MOD 102 males and 75 females in MARD
Time since diagnosis (years)	P = 4.95E-06 11.5 ± 8.9 in SIDD 6.2 ± 8.0 in SIRD 5.6 ± 9.2 in MOD 6.4 ± 8.1 in MARD

We checked for confounding of the above factors with our identified cluster specific proteins and metabolites. We have added two **Supplementary Tables 10 and 11** including this information. We have also checked the impact on the cluster-specific analysis after correcting for the different confounders. The p-values of the associations were only marginally weaker for most confounding factors (p-values remained between 1×10^{-4} - 1×10^{-2}). We have added a paragraph to the manuscript describing the effect of confounding on cluster membership.

“Assessment of the effect of confounding on cluster membership and individual metabolites and proteins.

*We tested for the association of the following confounding factors: sex, time since diagnosis, fasting time, medication (lipid lowering, diabetes, and hypertension drugs), and T2D polygenic risk score with the cluster assignment. The score for type 2 diabetes was computed using variants and weights from a previous study [Khera, A. V. et al, Nature Genetics 2018] based on summary statistics from a recent GWAS in patients with type 2 diabetes. After correcting the five clustering variables for the significant confounders (sex, time since diagnosis, diabetes drugs, and hypertension drugs) all associations remained significant ($p < 0.05$). We further independently reported potential confounding of the various factors with the identified cluster specific proteins and metabolites (**Supplementary Table 10** and **Supplementary Table 11**).”*

Little mention is made of effects of **diet and microbiome** on cluster outcomes and how this may be influencing both BMI, insulin resistance and overall tendency to develop diabetes. In addition, both are likely to affect population tendencies to develop diabetes and the metabolites later measured.

Response:

*Thank you, we have now added an analysis of diet- and microbiome-related metabolites. For this purpose, we have referred to a recent study [Noam Bar et al. Nature 2020] which identified 335 metabolites that were significantly explained by diet and 182 metabolites that were affected by the microbiome. Out of the 35 metabolites that overlapped with our identified cluster specific metabolites, no metabolites had a strong correlation with microbiome ($R^2 < 0.15$) and only three metabolites had a weak correlation with diet (X – 11315; $R^2 = 0.34$, 1,5-anhydroglucitol (1,5-AG); $R^2 = 0.27$, and methyl glucopyranoside (alpha + beta); $R^2 = 0.24$). We have added **Supplementary Table 12** where all the correlations between cluster specific metabolites and diet/microbiome are listed.*

We have added the following text to the manuscript:

*“We further tested for the effects of diet and microbiome on cluster outcomes. Using data from a previous study [Noam Bar et al. Nature 2020] which identified 335 metabolites that were significantly explained by diet and 182 by the microbiome. We extracted the explained variance of diet and microbiome (R^2) by our cluster specific metabolites (**Supplementary Table 12**). However, we found no strong correlation between these metabolites and microbiome ($R^2 < 0.15$) nor diet ($R^2 < 0.35$).”*

I would also draw the authors' attention to an article which came out late last year on a similar theme, all-be-it in a different population. Roderick C. et al. Distinct Molecular Signatures of Clinical Clusters in People With Type 2 Diabetes: An **IMI-RHAPSODY** Study. *Diabetes* 1 November 2021; 70 (11): 2683–2693. <https://doi.org/10.2337/db20-1281>. They may be interested in comparing their findings.

Response:

We thank the reviewer for pointing us towards this very interesting paper. We now integrate the results of this paper together with the new results from the AGES study to the replication part of our paper. However, please note that there are several differences between the studies which complicate direct comparison between the two. We attempted to test for general coherence between the two studies where appropriate, and added a section to the manuscript detailing this:

*“Slieker et al. also analyzed the molecular signatures of the diabetes subtypes after obtaining metabolomic, lipidomic, and proteomic data from plasma. Molecular measures were measured using ultrahigh-performance liquid chromatography-tandem mass spectrometry (UHLPC-MS/MS) for metabolomics (M=15), and the SomaLogic somascan platform for proteomics (P=1,195). We tested the 906 protein measures reported to be specific for the SIDD, SIRD, MOD, and MARD clusters, and found that 38%, 2%, 19%, and 33% of the proteins replicated respectively, while 75%, 50%, 57%, 66% of the proteins had consistent directionality, respectively (**Supplementary Table 14**).”*

....

*“We further attempted to replicate 14 of the 15 measured metabolic associations reported in Slieker et al. (measured using UHLPC-MS/MS). 50%, 25%, 0%, and 67% of the metabolites replicated in the SIDD, SIRD, MOD, and MARD clusters respectively, while 100%, 88%, 0%, and 100% of the metabolites had consistent directionality, respectively (**Supplementary Table 16**).”*

Specific points

Line 115: I would specify physician diagnosis of diabetes mellitus to distinguish it from **the much rarer diabetes insipidus** (which I have assumed was excluded from this study of diabetes)

Response:

The reviewer is correct - diabetes insipidus is a rare disorder and does not fall within our definition of T2D, which we define in the manuscript as follows:

“T2D was defined if any one of the following four conditions was met: first, having a physician diagnosis of diabetes based on the questionnaire (13.3% of all participants), second, being under diabetes treatment based on the QBB questionnaire (11.6%), third, having an HbA1c > 6.5% (10.5%), or fourth, having a random glucose level > 200 mg/dL (3.3%). Based on this definition,

15.4% of individuals were defined as T2D cases. Most individuals with a physician diagnosis of diabetes were on oral anti-diabetic treatment (73.7%), insulin treatment (23.9%), diet treatment (38.3%) and/or physical activity (16.2%) (see Supplementary Figure 1 for a Venn diagram)."

Line 165: **was it random which patients had phenotype and metabolomics/proteomics data?**
Or was this in any way biased.

Response:

To clarify this point, we have added the following text to the Methods:

"Samples selected for metabolomics and proteomics measurements correspond to the first ~3,000 participants of the Qatar Biobank. No specific selection criterion was applied."

Line 168: please clarify in the text if you have given C.Vs as a percentage (0.172%) or as a decimal i.e., 17%

Response:

All C.Vs are now given as decimal, i.e., 17%

Line 170 onwards: Nothing was said here about any **quality assurance of the metabolomics datasets**. For example, were any quality control measures applied to the metabolomics data and were any compounds excluded from further analysis due to poor technical reproducibility? How were **batch differences accounted for in measurements of proteomics and metabolomics datasets**, especially given the semi-quantitative nature of most of the measurements.

Response:

We carried out appropriate QC measures on both the proteomics and metabolomics datasets. In addition, the fact that most of the case-control diabetes-omics associations replicate, provides further validation regarding the quality of the data.

The metabolomics data was acquired on the Metabolon HD4 platform and implemented and run by the Antidoping Laboratory Qatar. We have previously described these measurements in detail in Suhre et al. Metabolites 2022 and have accordingly cited it but not repeated the details in the current paper.

Briefly: quality assurance is a central part of Metabolon's protocol and is based on measuring pooled customer samples, endogenous standards, and blanks at regular intervals. Samples were measured in batches of 36. These batches were randomized on sex, age, body mass index, diabetes state, prevalent hypertension, and HbA1c in such a way that none of these parameters

associated with run day. Raw data was transferred to Metabolon and processed following its in-house protocol for peak identification and quantification. This step also includes Metabolon's standard approach of median normalization between batches. Metabolites of low quality would be excluded before delivery to the client. We used the data as provided by the Qatar Biobank and only excluded metabolites with a degree of missingness >x% entirely from the analysis.

The proteomics data was measured on the Somalogic platform at Weill Cornell Medicine in Qatar. Like the Metabolon platform, protocols provided by the company were strictly followed and all QC steps that apply to the generation of Somalogic data were followed, i.e., the raw data was transferred to Somalogic and processes and QC'ed according to the company's standards. Normalization between batches is part of this process and is based on spiked in standards.

In summary, the study data has been acquired by the Qatar Biobank using platforms from two leading metabolomics and proteomics service providers, Metabolon and Somalogic, following their previously published protocols. The overall validity of their approaches has been shown especially in several genome-wide association studies. We report the results of the QC from both platforms in our paper as follows:

"Metabolon: Instrument variability, based on measurement of internal standards, was 12% and total process variability, based on endogenous biochemicals measured in repeated reference samples, was 16%."

"Somalogic: Quality control was performed using repeated measures of two QC samples. The median coefficient of variance (CV) was 7.3% for both QC samples, based on 51 and 54 repeated measures, respectively. 95% of the aptamers had a CV below 17.2% and 17.6%, respectively, and 5% had a CV below 4.6% and 4.1%, respectively."

To clarify the reviewer's question, we added:

"Data was used as provided by Somalogic and Metabolon and includes their respective in-house batch normalization steps. No additional QC was applied that led to exclusion of samples. Metabolites with more than 90% missingness were excluded. No proteins were excluded."

Line 251: Just a comment: you used **Bonferroni to correct for multiple testing**. This is not wrong, but it may be overly strict, especially given the inherent co-dependency of many of the metabolites.

Response:

We agree with the reviewer that Bonferroni correction may be overly strict for correlated traits. However, by focusing on the strongest associations we not only assure a low false positive rate, but also select for associations with the largest effect sizes, which we assume may be the most relevant ones clinically.

P11 Line 285: reference is made to the cluster sizes being broadly comparable between QBB and ANDIS. The authors mention a **difference in the SIRD group sizes; do they have an explanation for this?** Could it be related to the younger age of onset? Or the cut off values used for the clustering?

Response:

We agree the one central difference between QBB and ANDIS (and the other studies we compare to) is the younger age of the QBB cohort. This is not a feature of the cohort, but of the general population in Qatar. Despite this, Qatar has one of the highest diabetes rates worldwide. We believe these differences are specific to the population and are the most intriguing when comparing the ANDIS and QBB profiles (see Figure 2. in the manuscript below), and we highlight this in the discussion:

“Our cluster sizes are comparable between QBB and ANDIS for the most part. The difference in the SIRD group size could be due to the younger age of diabetes onset in the QBB cohort, which reflects upon the centers of clusters and ultimately the distance of samples from cluster centers. The mean age in the SIRD cluster in ANDIS is between 50-75 compared to between 30-40 in QBB. Another difference between the studies, is that most patients with type 2 diabetes in ANDIS were above 50 years old at the time of diagnosis, while most patients with type 2 diabetes in the QBB cohort were between 30-50 years of age at diagnosis. However, the BMI trends between ANDIS and QBB were similar, irrespective of the younger age of the QBB patients.”

Figure 2. Cluster characteristics and cluster distribution in QBB. K-means clusters were derived using the QBB training set and classification was applied to the testing set using the training set cluster coordinates. **A)** Distributions of HbA_{1c}, BMI, age, HOMA2-B, and HOMA2-IR are shown for each cluster in QBB and ANDIS. HbA_{1c}, BMI, HOMA2-B, and HOMA2-IR all followed the same trend in QBB and ANDIS, but individuals in the MOD cluster were younger than the other clusters in QBB.

P298: the authors mention that clustering male and female participants separately showed little difference in eventual cluster membership, but **was there any male/female difference in cluster membership**, or the average cluster values? I was particularly interested in whether **individuals on the borderline of different cluster groups** were more likely to be one sex, and whether **menopause** was a confounding factor in cluster membership. Do males and females tend to have different **outcomes** in their clusters?

Response:

*The male and female cluster center values are included in **Supplementary Table 4**. We have also added an additional plot (**Supplementary Figure 5**) showing the average cluster values stratified by gender. The average cluster values for the clinical variables were comparable between males and females. In **Supplementary Figure 4B**, we clustered males and females together and then separately. We observed a 7% change in the cluster assignments when clustering the full data set compared to clustering the gender specific datasets. We added the following text to the results:*

“Although there was a slight imbalance between males and females in the clusters (51 vs. 42 in SIDD, 11 vs. 6 in SIRD, 23 vs. 66 in MOD, and 102 vs. 75 in MARD), clustering the males and females separately resulted in little difference in the eventual cluster membership. The 26 (7%) individuals on the borderline of the different cluster groups included 11 males and 15 females. Due to such small numbers, it is difficult to judge whether the individuals on the borderline of the different cluster groups were more likely to be one sex.”

Menopause data is not available in QBB and hence could not be tested for confounding the cluster membership. Finally, regarding the last point, this study is a cross-sectional study, so we do not have outcome data.

Supplementary Figure 5. Average cluster values stratified by gender. The average cluster values for the five clinical variables (HbA1c, Age, BMI, HOMA2-B, and HOMA2-IR) are comparable between males and females.

Table 1: please clarify is the percentage in the exercise groups are cumulative or separate i.e., were individuals who exercised over 3 hours a week counted in both groups.

Response:

The percentages are cumulative. We have clarified this in the table header by adding:

“The second physical activity question is cumulative and includes the individuals from the first physical activity question.”

Time since disease was diagnosed would also be nice to include here – and if possible, an analysis to confirm that different classifications are genuinely separate diagnoses and not marking a linear progression of diabetes.

Response: We have now added the time since disease diagnosis in our sensitivity analysis for confounders (please refer to previous comment). The time since diagnosis in years is associated with the cluster membership in a linear model ($P = 4.95 \times 10^{-6}$; 11.5 ± 8.9 in SIDD, 6.2 ± 8.0 in SIRD, 5.6 ± 9.2 in MOD, and 6.4 ± 8.1 in MARD). We have added the following text to the discussion:

“It appears that the time since disease diagnosis is higher in SIDD compared to the other subtypes, suggesting that the more severe subtypes of type 2 diabetes may possibly be the result of disease progression. However, it is also reasonable that the more severe subtypes have been diagnosed for a longer period.”

Several supplementary tables were referenced in the main paper but no supplementary tables were available in the copy I had to review.

Response:

We checked the source zipfile that was provided to the reviewers and the supplementary tables were included. We are not sure what went wrong in this case, but hopefully they are included with the current revision. Please let the editor know, if that is not the case.

Line 404: How certain are the authors that the plasma level of **phosphate** was comparable to the in-vivo measurement and not a post-collection artefact?

Response:

There is a strong correlation between the phosphate measurement from Metabolon and that measured by clinical biochemistry, as shown in the figure eliminating the possibility of potential post-collection artifacts.

Line 468 onwards: the authors have mentioned specific **medication** differences between clusters, but were these checked for their **confounding effects** on the eventual results, especially since some of the medications would be immunomodulatory.

Response:

We addressed the confounding effects, including medications like lipid lowering drugs, diabetes drugs, hypertension drugs, on the cluster membership in a previous comment. We repeat the text that we added to the manuscript here:

“Assessment of the effect of confounding on cluster membership and individual metabolites and proteins.

*We tested for the association of the following confounding factors: sex, time since diagnosis, fasting time, medication (lipid lowering, diabetes, and hypertension drugs), and T2D polygenic risk score with the cluster assignment. The score for type 2 diabetes was computed using variants and weights from a previous study [Khera, A. V. et al, Nature Genetics 2018], which is based on summary statistics from a recent GWAS in patients with type 2 diabetes. After correcting the five clustering variables for the significant confounders (sex, time since diagnosis, diabetes drugs, and hypertension drugs) all associations remained significant ($p < 0.05$). We further independently reported potential confounding of the various factors with the identified cluster specific proteins and metabolites (**Supplementary Table 10 and Supplementary Table 11**).”*

Line 518: it was not clear to me how you had made the link between association of certain **immunomodulatory proteins and their role in a feedback resolving inflammation**.

Response:

We added this text to the discussion:

“The effects of immunomodulatory proteins on the immune system can be associated with different chronic inflammatory diseases such as diabetes, obesity, cardiovascular diseases, and cancer. For example, elevations in C2 indicate upregulation of the complement pathway in T2D, which could be a response to hypoglycemia. Hypoglycemia induces pro-inflammatory proteins such as interleukins [Kitabchi et al., Metabolism 2009]. This upregulation of the complement pathway suggests that both intrinsic and alternative pathway activation may be driven by the underlying inflammation in T2D. An inflammatory response can be rapidly induced in response to hypoglycemia and would likely drive other acute response pathways, such as the complement cascade, with significant crosstalk between the two [Leslie et al., Science 2012].”

Reviewer #3 (Remarks to the Author):

This study extended the Ahlqvist et al. diabetes subtype findings to a large Arabic cohort by 1) (somewhat) confirming similar clinical diabetes subtypes by using both de novo clustering methods in a training dataset of 2,740 individuals that were then evaluated in the testing set of 2,155 and 2) leveraging metabolomics and proteomics to profile circulating biomarkers associated with these subtypes in the testing cohort of 2,155 individuals. The SOMAscan 1,300 plex proteomics platform and Metabolon metabolomics platforms were used. Strengths of the study include the **novelty of evaluating the Ahlqvist et al. diabetes subclassification system in a new population** with high T2D burden, and **extending the phenotyping to metabolomic and proteomic profiling** to identify circulating biomarkers associated with this novel diabetes classification system. One **significant limitation was that fact that was a lot of variability in the time fasting prior to blood collection** (with 23% having not fasted) and that only 50% fasted for 8 hours which can significantly affect metabolite—and to a possibly lesser extent protein concentrations. Another **major concern was the small numbers** considering that these metabolite and protein associations were determined based on 420 individuals with diabetes that were subdivided into 4 classes (with one subclass only containing 17 individuals).

Response:

Thank you for identifying the strengths of our study. Regarding the number of individuals in the study, while we appreciate that other studies may have a larger sample size, our study of 420 individuals with diabetes and extensive metabolomics (M=1,159) and proteomics (P=1,305), is the first study to have such deep molecular metabolomic profiling, and more importantly it is the first

study to be carried out in an Arab cohort. Therefore, our novelty lies in both applying the Ahlqvist classification in a new population and extending our understanding of the underlying etiology of the different subtypes in this classification with deep molecular profiling. In addition, we have only reported the Bonferroni significant associations, which whilst overly conservative, risking the loss of interesting findings, provides a certain degree of confidence in relation to interpretation despite the limited sample size.

Regarding the variability in the time fasting prior to blood collection, although this was already addressed in our manuscript, we have carried out an additional sensitivity analysis to determine the impact of the variability in fasting time on the obtained clusters. The association between the fasting time and the cluster membership was not significant ($P = 0.08147$). We have added this to the text in addition to the text below.

“HOMA2-IR and HOMA2-B estimates are based on plasma glucose and C-peptide (Cpep) levels and are sensitive to the fasting state. Other studies have reported on clustering using non-fasting values by using HDL-cholesterol and C-peptide, which is a proxy for insulin resistance. As QBB participants were not all in a fully fasted state (77% of the individuals had fasted for over two hours at the time of enrolment and 50.7% had fasted for over 8 hours), we tested the sensitivity of the cluster assignments to self-reported time-since-last-meal. Using linear regression, we estimated fasting HOMA2-IR and HOMA2-B and used the corrected values for clustering. 98% of the cluster assignments remained unchanged, indicating that the clustering is robust to fasting state (**Supplementary Figure 4C**).”

When clustering using uncorrected vs. corrected HOMA variables, most individuals maintained the same cluster assignment using the fasting time - corrected data.

Major (detailed) Critiques:

1. Why was BMI adjustment only completed for the metabolomics analyses and not for the proteomics analyses? The rationale provided in the manuscript was not sufficient.

Response:

We carried out BMI adjustment only in the metabolomics analyses and not for the proteomics analyses for consistency with previous work [Gudmundsdottir et al. Diabetes 2020]. We have now added a sensitivity analysis where we added BMI in the proteomics analysis. We observe no significant differences in the effect size when comparing the analyses with and without BMI.

2. Please clarify the exact methods used for the weighting in each cohort. This was quite unclear: “After using the ANDIS coordinates instead of the QBB coordinates to classify QBB patients, 35% changes in the cluster assignments were observed.” Please explain.

Response:

We apologize for the lack of clarity and have modified the text as follows:

“The ANDIS coordinates were used instead of the QBB coordinates to classify QBB patients. Gender specific type 2 diabetes cluster centers (SIDD, SIRD, MOD, and MARD) from Ahlqvist et al. were obtained. After computing the Euclidean distance between the four clusters and each individual in QBB, each individual was assigned to the cluster with the shortest distance. When comparing the cluster assignment that was based on the QBB coordinates vs. ANDIS coordinates, a 35% change in the cluster assignments was observed.”

3. Given the **variability in fasting status** prior to blood sample collection, sensitivity analyses were conducted for the diabetes subtype clustering. However, were sensitivity analyses conducted for the metabolomics and proteomics analyses? **Fasting** is known to influence metabolite levels, and possibly to a lesser degree, protein levels.

Response:

We found little impact of the variability in fasting status on the cluster assignment as the association between fasting time and the cluster membership was not significant ($p=0.08147$). This has now been added to the manuscript:

“Assessment of the effect of confounding on cluster membership and individual metabolites and proteins.

We tested for the association of the following confounding factors: sex, time since diagnosis, fasting time, medication (lipid lowering, diabetes, and hypertension drugs), and T2D polygenic risk score with the cluster assignment. The score for type 2 diabetes was computed using variants and weights from a previous study [Khera, A. V. et al, Nature Genetics 2018], which is based on summary statistics from a recent GWAS in patients with type 2 diabetes. After correcting the five clustering variables for the significant confounders (sex, time since diagnosis, diabetes drugs, and hypertension drugs) all associations remained significant ($p<0.05$). We further independently reported potential confounding of the various factors with the identified cluster specific proteins and metabolites (**Supplementary Table 10 and Supplementary Table 11**”).”

4. Given the technical **differences between the SOMAscan and Olink** platforms, what was done to align the relative aptamer/protein concentrations between QBB and ANDIS?

Response:

We agree that measurements of the same protein between Somalogic and Olink can vary. In addition to our replication using the Somascan platform, we also included a replication using the Olink platform (**Supplementary Table 15**).

5. **Very few metabolites validate on the Olink platform.** What were the correlations between the proteins measured on both platforms if there were any samples run on both platforms? And more generally, can the authors **supply a table to inform us what data support the specificity of the platform(s)**?

Response: Concordance and specificity of the Olink and Somalogic platforms is an ongoing matter of discussion. No samples were run on both platforms. A recent paper, [Lopez-Silva et al. Clin J Am Soc Nephrol; 2022] provides such a comparison. We have now added a replication with the same (Somalogic) platform. To address this issue, we have added the following sentence to the discussion:

“Concordance and specificity of the Olink and Somalogic platforms is an ongoing matter of discussion [Lopez-Silva et al. Clin J Am Soc Nephrol; 2022]. Some of the proteins that were not replicated on the Olink platform (ANDIS) could be a result of differences in aptamer and antibody binding.”

6. I struggle to understand the significance of Table 2.

Response:

*We agree that information in **Table 2** is only of descriptive value as it characterizes the study population's general medication. We have moved it to the Supplementary Tables.*

7. In Figure 6, how were these proteins and metabolites selected? What additional information does this figure provide compared to Figure 4 and 5?

Response:

*These figures are more detailed plots of the data provided in **Figures 4 and 5** for five traits that were selected as examples of the most distinctive cluster specific metabolites or proteins. The additional information provided is the full box plot style presentations of the entire data, with whiskers and individual points for outliers, while **Figures 4 and 5** only show the mean and confidence intervals. We included these examples with the intent to introduce the reader to **Supplementary Figure 7 and Supplementary Figure 8**, which contain similar plots for all traits shown in **Figures 4 and 5**. We believe that these figures are important as they allow the reader to verify that associations are not driven by outliers or skewed distributions.*

8. **Please clarify in the limitations section** why the measurement of relative abundance of proteins and metabolites are “not a concern in this kind of study.”

Response:

Thank you, we have clarified this as follows:

“The deep molecular phenotyping using the SOMAscan and Metabolon technologies provided only relative abundances of protein and metabolite levels respectively, not absolute concentrations. However, this is not a concern as the association statistics we use, i.e., linear models and t-tests, are invariant under scaling and translation of the data.”

9. The **variability in hours fasted prior to blood draws** should be discussed in the limitations given its relevance especially to the metabolomics analysis.

Response:

The association between fasting and cluster membership was not significant ($p=0.08147$). We have added a section in the manuscript and additional supplementary tables to describe the impact of various confounders, including fasting time. Our new supplementary table showing the

association between the cluster-specific metabolites and different confounders, including fasting time, can be used to flag the specific metabolites which should be interpreted with caution.

The following has been added to the manuscript:

“Assessment of the effect of confounding on cluster membership and individual metabolites and proteins.

*We tested for the association of the following confounding factors: sex, time since diagnosis, fasting time, medication (lipid lowering, diabetes, and hypertension drugs), and T2D polygenic risk score with the cluster assignment. The score for type 2 diabetes was computed using variants and weights from a previous study [Khera, A. V. et al, Nature Genetics 2018], which is based on summary statistics from a recent GWAS in patients with type 2 diabetes. After correcting the five clustering variables for the significant confounders (sex, time since diagnosis, diabetes drugs, and hypertension drugs) all associations remained significant ($p < 0.05$). We further independently reported potential confounding of the various factors with the identified cluster specific proteins and metabolites (**Supplementary Table 10 and Supplementary Table 11**).”*

10. Why do the authors exclude **pre-diabetes**?

Response:

The clustering approach is designed for patients with diabetes, i.e., one key variable that enters the scheme is “Age at diagnosis”. Hence, it cannot be applied to individuals with pre-diabetes. We agree that using a similar approach to pre-diabetes is of interest, and metabolomics and proteomics data could play a guiding role. However, we believe that this question is beyond the scope of the present investigation.

11. **Why was the metabolomics replication so poor?** “... 41 (23%) of these associations were statistically significant at a Bonferroni level ($p < 0.05/175$)”. Isn’t that a major concern?

Response:

The replication cohort used for the metabolomics study had less power than the discovery study. Therefore, many associations did not reach the required significance level. However, all associations between QBB and QMDiab were directionally concordant. There is a wealth of other studies on metabolomics-diabetes associations, in which we found consistent signals with our metabolomics-diabetes associations. However, having a full-fledge comparison of metabolomics-diabetes associations with all available literature is beyond the scope of our current study. The primary purpose for this analysis was to validate the reliability of metabolomics measurements.

To make this clearer we now state:

“All associations were directionally concordant. Despite being less powered, 41 (23%) of these associations were statistically significant at a Bonferroni level ($p < 0.05/175$) in the QMDiab study.”

12. It is still not clear to me **how much of the biochemical differences are due to the underlying med use**...much less BMI or other clinical traits driving the subclasses.

Response:

We have added two new Supplementary tables showing the association of the identified cluster-specific proteins/metabolites with various confounders i.e., sex, time since diagnosis, fasting time, diabetes genetic score, smoking, lipid lowering drugs, diabetes drugs, and hypertension drugs. Although some significant confounding effects of the medication use contribute to the cluster membership, all metabolite/protein to cluster associations remained significant ($p < 0.05$) after correction of the clustering variables for these confounders. Please refer to our previous responses addressing confounding of medication usage.

Minor Critiques:

1. Please **define and briefly describe the ANDIS trial** for readers who are not aware this is the population that the original Ahlqvist et al. diabetes clustering was derived from.

Response:

We have added a section to the Methods defining and briefly describing the ANDIS trial:

“ANDIS study

The ANDIS study included patients with newly diagnosed diabetes ($n=8,980$) from the Swedish All New Diabetics in Scania cohort. This cohort was used in the original study that first defined the T2D subtypes. Data-driven k-means cluster analysis using six variables (glutamate decarboxylases antibodies, age at diagnosis, BMI, HbA1c, and HOMA2-B, and HOMA2-IR, was carried out on these patients. Five distinct clusters of diabetes subtypes with significantly different patient characteristics and risk of complications were identified. These subtypes included one autoimmune or severe autoimmune diabetes (SAID), and four subtypes of type 2 diabetes, namely severe insulin dependent (SIDD), severe insulin resistant (SIRD), mild obesity-related (MOD), and mild age-related (MARD).”

In summary, the **assignment of individuals into these various sub-classes will be particularly useful** if 1) they **respond differently to interventions** or 2) if the **metabolomics/proteomics inform causative biology**. The second point is the focus of this paper, but it seems early as one reads through the discussion. The associations seem to just be downstream of the clinical features that distinguish the groups (age, BMI, etc....)

Response: We agree with the summary and believe that our findings are important given that we have translated the diabetes clustering scheme for the first time to an Arab population and applied broad as well as deep metabolomics and proteomics measurements to characterize these clusters. Determining which of the clinical traits are downstream of the features remains to be investigated and is potentially of great interest to drug development. Answering this question requires the use of tools like Mendelian randomization in much larger studies.

Reviewer #4 (Remarks to the Author):

Zaghlool et al. show that previously established type 2 diabetes subtypes derived from Swedish data replicate in an Arab population. These subtypes have been replicated in various other populations as well, but **now for the first time also in Arabs**. Further, the authors analyzed metabolic and proteomic data in relation to these subtypes and provide insights into the underlying etiology of these subtypes. While the **study is well-conducted and the findings are supported by partial replication in other studies and literature**, I would like to see **more discussion and better presentation of the novel aspect** of this study regarding the metabolic and proteomic signatures. Below I detail my comments.

Response:

We thank the reviewer for their appreciation of our work. As requested in the revised version, we have included more discussion and (hopefully) a better presentation of the novel aspect of our study, including extensive sensitivity analyses and other modifications made in response to the other reviewers' comments.

1. It seems that replication of T2D-associated proteins and metabolites was attempted in AGES and QMDIAB but cluster-specific proteins were replicated in ANDIS. Is there a reason why the clustering and cluster-specific associations were not attempted in AGES and QMDIAB?

Response:

For this revision, we have now added a replication of the protein cluster-specific associations in AGES. The betas were consistent for most proteins (See figure). The following text was added to the manuscript:

*“We attempted replication of T2D subtype specific protein associations in a different population using the same proteomics platform. This was attempted for all 47 subtype specific protein associations (**Supplementary Table 13; Supplementary Figure 12**) using proteomics measurements from the Somalogic platform in the AGES study. Data was available for 588 individuals (SIDD = 61, SIRD = 84, MOD = 120, MARD = 212). Here, 18 protein associations were replicated after accounting for multiple testing ($p < 0.00106$; $0.05/47$). Additionally, 16 protein associations were nominally significant ($p < 0.05$) and directionally consistent in AGES.*

Furthermore, 96% of the associations were directionally consistent between the two cohorts (Supplementary Figure 13).”

Supplementary Figure 12. AGES study replication of proteins that distinguish individual diabetes subtypes in QBB. The dots and bars represent the mean protein values and the 95% confidence intervals of the means for proteins that are different in one of the four T2D subtypes compared to all others. Values are normalized by the mean of the respective reference subtype. Data for SAID and the control group are shown for reference.

Supplementary Figure 13. Comparison of regression coefficients for subtype-specific proteins between QBB and AGES. Most of the associations (96%) were directionally concordant.

Unfortunately, QMDiab does not provide HOMA2-B and HOMA2-IR measures, so that clusters could not be computed for this study. In addition, AGES has no metabolomics data.

2. Validation of clusters. Please **specify the parameters/tests that were used to assess the fit of the clusters in the test set.**

Response:

To assess the fit of the clusters in the test set, we applied the cluster coordinates obtained from the training set to the testing set and compared those cluster assignments to the cluster assignments in the case of deriving the cluster memberships solely from the testing set. There were minor differences as shown in the following Sankey diagram (**Figure 2B**).

We also compared the distributions of the five clinical variables (HbA1c, BMI, age, HOMA2-IR, HOMA-B) between the training and test sets. We found no significant difference ($p > 0.05$) in the five variables. This information is included in the demographics table (Supplementary Table 1).

We further compared the cluster centers, the average inter-cluster similarity, and the average intra-cluster similarity derived between the training and testing sets and noted less than 5% shift in the relative cluster parameters.

Figure 2B. The testing set clusters were similar to the training set clusters, regardless of whether they were assigned based on the training set coordinate centers or derived de novo for the testing set using K-means clustering. Minor changes in the cluster assignments (2%) were observed when clustering the data using the training set coordinates versus the testing set coordinates.

3. Why was a **different transformation for proteins (Box-Cox vs. log)** used in the replication data AGES? Did the authors check whether this would influence the results.

Response:

We used the Box-Cox transformation vs. log for the replication in AGES for compatibility with the previously published results [Gudmundsdottir et al., Diabetes; 2020]. To answer the reviewer’s question, we checked whether this would influence the results and only found a minor impact on the replication.

To clarify this point, we have added the following information to the paper:

“Variable transformations were chosen to be compatible with the replication cohorts (Box-Cox for AGES, log for QMDiab). We verified that using a log transformation did not substantially change the results.”

4. The clusters were **transferable to the Arab population but with population-specific coordinates** (Figure 2B and C). I think **this should be highlighted** better in the text (and abstract), especially regarding potential clinical utilization of the clusters.

Response: We agree with the reviewer and have highlighted this better in the abstract (refer to re-written abstract) and throughout the text in the manuscript (refer to tracked changes in the revised manuscript).

5. How would the authors take the results further? In the conclusions it is stated that the findings “have the potential to identify novel pathways involved in the development and progression of T2D complications, improve risk prediction, and enable more personalized treatment approaches”. I would hope to see a more detailed discussion and perhaps some specific figures on this aspect, which seems to me to be the most interesting part of the paper.

Response:

We have added the following to the discussion:

“By identifying novel pathways involved in the development and progression of T2D complications, these results can be taken forward by carrying out Mendelian randomization studies of the association of cluster specific metabolites and proteins with diabetes. This type of analysis has the potential to distinguish between potentially actionable therapeutic targets from those that are downstream of the disease and that could therefore serve as diagnostic biomarkers. This is a preliminary step to carrying out randomized control trials for drug testing. Our associations can also be used to generate hypotheses for follow-up studies on the processes that might lead to these associations.”

Minor comments:

1. Abstract. Currently it is unclear by only reading the abstract that the n relates to the number of metabolites/proteins rather than individuals.

Response:

We have now clarified this in the abstract:

“The subtype clustering approach was applied to 631 individuals with T2D from the Qatar Biobank (QBB) and validated in an independent set of 420 participants from the same population. Cluster-specific signatures of blood circulating metabolite (N=1,159) and protein levels (N=1,305) were established. “

2. Table 1 smoking: is shisha in second-hand smoking the same as water pipe in the current smoker?

Response:

Yes, it is the same. We replaced “shisha” with “water pipe” for consistency.

3. P.8 line 207: define ANDIS.

Response:

We defined it as the “Swedish All New Diabetics in Scania cohort”

4. P. 12 line 323: replicate “in” other populations

Response:

This was corrected.

5. P. 19 line 483: define MENA.

Response:

This has been changed to “Middle East and North Africa”.

6. P. 19 line 501: should it be HOMA2-IR?

Response:

This was corrected.

REVIEWERS' COMMENTS

Reviewer #1 (Remarks to the Author):

The authors responded to my comments in a very satisfactory manner.

Reviewer #2 (Remarks to the Author):

The authors have made considerable efforts to address my concerns in the review. I have nothing further to add.

Reviewer #3 (Remarks to the Author):

The authors have provided a rather robust response to all of the authors' queries. Many new analyses have been performed. The majority of the important issues have been addressed/clarified.

Reviewer #4 (Remarks to the Author):

The authors have addressed well my and other Reviewers' comments and I have no further comments.